# Separable neural signatures of confidence during perceptual decisions

**Tarryn Balsdon[1,2]\*, Pascal Mamassian[1†], Valentin Wyart[2†]**

[1]Laboratoire des Systèmes Perceptifs (CNRS UMR 8248), DEC, ENS, PSL University, Paris, France; [2]Laboratoire de Neurosciences Cognitives et Computationnelles (Inserm U960), DEC, ENS, PSL University, Paris, France

**Abstract** Perceptual confidence is an evaluation of the validity of perceptual decisions. While there is behavioural evidence that confidence evaluation differs from perceptual decision-making, disentangling these two processes remains a challenge at the neural level. Here, we examined the electrical brain activity of human participants in a protracted perceptual decision-making task where observers tend to commit to perceptual decisions early whilst continuing to monitor sensory evidence for evaluating confidence. Premature decision commitments were revealed by patterns of spectral power overlying motor cortex, followed by an attenuation of the neural representation of perceptual decision evidence. A distinct neural representation was associated with the computation of confidence, with sources localised in the superior parietal and orbitofrontal cortices. In agreement with a dissociation between perception and confidence, these neural resources were recruited even after observers committed to their perceptual decisions, and thus delineate an integral neural circuit for evaluating perceptual decision confidence.

## Introduction

Whilst perception typically feels effortless and automatic, it requires probabilistic inference to resolve the uncertain causes of essentially ambiguous sensory input (*Helmholtz, 1856*). Human observers are capable of discriminating which perceptual decisions are more likely to be correct using subjective feelings of confidence (*Pollack and Decker, 1958*). These feelings of perceptual confidence have been associated with metacognitive processes (*Fleming and Daw, 2017*) that enable self-monitoring for learning (*Veenman et al., 2004*) and communication (*Bahrami et al., 2012*; *Frith, 2012*). We are only just beginning to uncover the complex functional role of metacognition in human behaviour, and outline the computational and neural processes that enable metacognition. The study of perceptual confidence offers promising insight into metacognition, because one can use our detailed knowledge of perceptual processes to isolate factors which affect the computation of perceptual confidence.

At the computational level, perceptual decisions are described by sequential sampling processes (*Ratcliff, 1978*; *Vickers, 1970*), in which noisy samples of evidence are accumulated over time, until there is sufficient evidence to commit to a decision. The most relevant information for evaluating perceptual confidence is the quantity and quality of evidence used to make the perceptual decision (*Kepecs et al., 2008*; *Moreno-Bote, 2010*; *Vickers, 1979*). At the neural level, perceptual confidence could therefore follow a strictly serial circuit: Relying only on information computed by perceptual processes, with any additional processes contributing only to transform this information for building the confidence response required by the task. Indeed, confidence (or a non-human primate proxy for confidence) can be reliably predicted from the firing rates of neurons coding the perceptual decision itself (*Kiani and Shadlen, 2009*), suggesting that confidence may be a direct by-product of perceptual processing. However, a large body of behavioural studies suggest that the computation of confidence is not strictly serial. Confidence can integrate additional evidence after

\*For correspondence:
tarryn.balsdon@ens.fr

†These authors contributed equally to this work

the observer commits to their perceptual decision (*Pleskac and Busemeyer, 2010*; *Baranski and Petrusic, 1994*), and while this continued evidence accumulation could incorporate only perceptual information, it implies that confidence evaluation does not directly follow from perceptual decision commitment (and therefore involves at least partially dissociable neural processes).

There is also evidence that perceptual confidence can rely on separate (non-perceptual) sources of information, such as decision time (*Kiani et al., 2014*) and attentional cues (*Denison et al., 2018*). This suggests that the processes involved in the computation of perceptual confidence may not be reduced to the same processes as for the perceptual decision. Higher order theories of metacognition propose a framework in which specialised metacognitive resources could be recruited for computing confidence across all forms of decision-making (a general metacognitive mechanism). Indeed, there is some evidence that confidence precision is correlated across different cognitive tasks (such as memory and perception; *Mazancieux et al., 2020*), suggesting a common source of noise affecting the computation of confidence across tasks (on top of the sensory noise; *Bang et al., 2019*; *Shekhar and Rahnev, 2021*).

It is reasonable to expect that a general metacognitive mechanism relies on processing in higher order brain regions. Several experiments have linked modulations in confidence with activity in a variety of subregions of the prefrontal cortex (including the orbitofrontal cortex, *Lak et al., 2014*; *Masset et al., 2020*; right frontopolar cortex, *Yokoyama et al., 2010*; rostro-lateral prefrontal cortex, *Fleming et al., 2012*; *Geurts et al., 2021*; *Cortese et al., 2016*; see also *Vaccaro and Fleming, 2018*, for a meta-analysis). Moreover, disrupting the processing in subregions of the prefrontal cortex (*Fleming et al., 2014*; *Lak et al., 2014*; *Rounis et al., 2010*) tends to impair (though not obliterate) the ability to appropriately adjust behavioural confidence responses, whilst leaving perceptual decision accuracy largely unaffected (although these results can be difficult to replicate, *Bor et al., 2017*; *Lapate et al., 2020*, and may not generalise to metacognition for memory; *Fleming et al., 2014*). A challenge in this literature is in specifically relating the neural processing to the computation of confidence, as opposed to transforming confidence into a behavioural response, or a downstream effect of confidence, such as the positive valence (and sometimes reward expectation) accompanying correct decisions. Moreover, identifying how these neural mechanisms could be separable from the underlying perceptual processes is important for understanding the computational architecture of metacognition.

One promising avenue of research for separating the mechanisms of metacognition from perceptual processes has been to utilise tasks where the observer may integrate additional evidence for confidence after they have committed to their perceptual decision (*Fleming et al., 2018*; *Murphy et al., 2015*), which presumably relies on processing independent of the perceptual decision. These studies show that post-decisional changes in confidence magnitude correlate with signals from the posterior medial frontal cortex. However, these signals could reflect processes occurring downstream of confidence, such as an emotional response to the error signal, which has been shown to drive medial frontal activity more strongly than decision accuracy (*Gehring and Willoughby, 2002*). Further research is therefore required to link neural processes specifically with the computation of perceptual confidence.

In this experiment, we aim to identify the neural processes specifically contributing to the computation of confidence, in a paradigm in which these processes can be delineated from those of perceptual decision-making. We exploit a protracted decision-making task in which the evidence presented to the observer can be carefully controlled. On each trial, the observer is presented with a sequence of visual stimuli, oriented Gabor patches, which offer a specific amount of evidence towards the perceptual decision. The orientations are sampled from one of two overlapping circular Gaussian distributions, and the observer is asked to categorise which distribution the orientations were sampled from. We manipulate the amount of evidence presented such that the observer tends to covertly commit to their perceptual decision before evidence presentation has finished, whilst continuing to monitor ongoing evidence for assessing their confidence (*Balsdon et al., 2020*). These covert decisions are evident from behaviour and computational modelling, and we show similarities between the neural processes of decision-making across conditions of immediate and delayed response execution.

To examine the computation of confidence, we compare human behaviour to an optimal observer who perfectly accumulates all the presented evidence for perceptual decisions and confidence evaluation. The optimal observer must accurately encode the stimulus orientation, the decision update

relevant for the categorisation, and add this to the accumulated evidence for making the perceptual decision. We uncover dynamic neural representations of these variables using model-based electro-encephalography (EEG), and examine how the precision of these representations fluctuate with behavioural precision. We find two distinct representations of the accumulated evidence. The first one reflects the internal evidence used to make perceptual decisions. The second representation reflects the internal evidence used to make confidence evaluations (separably from the perceptual evidence), and is localised to the superior parietal and orbitofrontal cortices. Whilst the perceptual representation is attenuated following covert decisions, the confidence representation continues to reflect evidence accumulation. This is consistent with a neural circuit that can be recruited for confidence evaluation independently of perceptual processes, providing empirical evidence for the theoretical dissociation between perception and confidence.

## Results

### Preview

We present analyses to address two key hypotheses in this experiment: First, that observers are prematurely committing to their perceptual decisions whilst continuing to monitor additional evidence for evaluating their confidence. And second, that there are separable neural signatures of the evaluation of confidence during perceptual decision-making. To address the first hypothesis, we use a combination of behavioural analyses and computational modelling, and in addition, show that the EEG signatures of response preparation are triggered from the time of decision commitment, even when this occurs seconds prior to the response cue. To address the second hypothesis, we use the stimulus evoked responses in EEG to trace the representation of the presented evidence throughout each trial. We show that these neural representations of the optimal accumulated decision evidence are less precise when the observers' behavioural responses were also less precise relative to optimal. We use this to isolate clusters of activity that specifically reflect the internal evidence used for observers' confidence evaluations beyond the presented evidence. We then localise the sources of this activity, and relate these processes back to observers' eventual confidence ratings.

### The computational architecture of perceptual confidence

Human observers (N = 20) performed two versions of the task whilst EEG was recorded. Across the two tasks, 100 predefined sequences of oriented Gabors were repeated for each observer, with stimuli presented as described in *Figure 1a*. In the Free task, the sequence continued until observers entered their perceptual decision (*Figure 1b*), indicating which category (*Figure 1d*) they thought the orientations were sampled from. Observers were instructed to enter their response as soon as they 'felt ready', on three repeats of each predefined sequence (300 trials in total). In the Replay task (*Figure 1c*), observers were shown a specific number of samples and could only enter their response after the response cue. After entering their perceptual decision, they made a confidence evaluation, how confident they were that their perceptual decision was correct, on a four-point scale. Importantly, the number of samples shown in the Replay task was manipulated relative to the Free task, in three intermixed conditions: in the Less condition, they were shown two fewer than the minimum they had chosen to respond to over the three repeats of that predefined sequence in the Free task; in the Same condition, they were shown the median number of samples; and in the More condition, four more than the maximum. The variability across repeats in the Free task means that in the More condition, observers were show at least four additional stimuli, but often more than that. There is an optimal way to perform this task, in the sense of maximising perceptual decision accuracy across trials. The optimal computation takes as decision evidence the log probability of each orientation given the category distributions (*Figure 1d*) and accumulates the difference in this evidence for each category (*Figure 1e*, *Drugowitsch et al., 2016*). We refer to the accumulated difference in log probabilities as the optimal presented evidence, $L$. Human observers may have a suboptimal representation of this evidence, $L*$, and we estimate the contribution of different types of suboptimalites (specifically, inference noise, and a temporal integration bias) with the help of a computational model (full details in Materials and methods and Appendix 1).

Based on our previous findings (*Balsdon et al., 2020*), we expected observers to prematurely commit to perceptual decisions in the More condition, whilst continuing to monitor sensory evidence

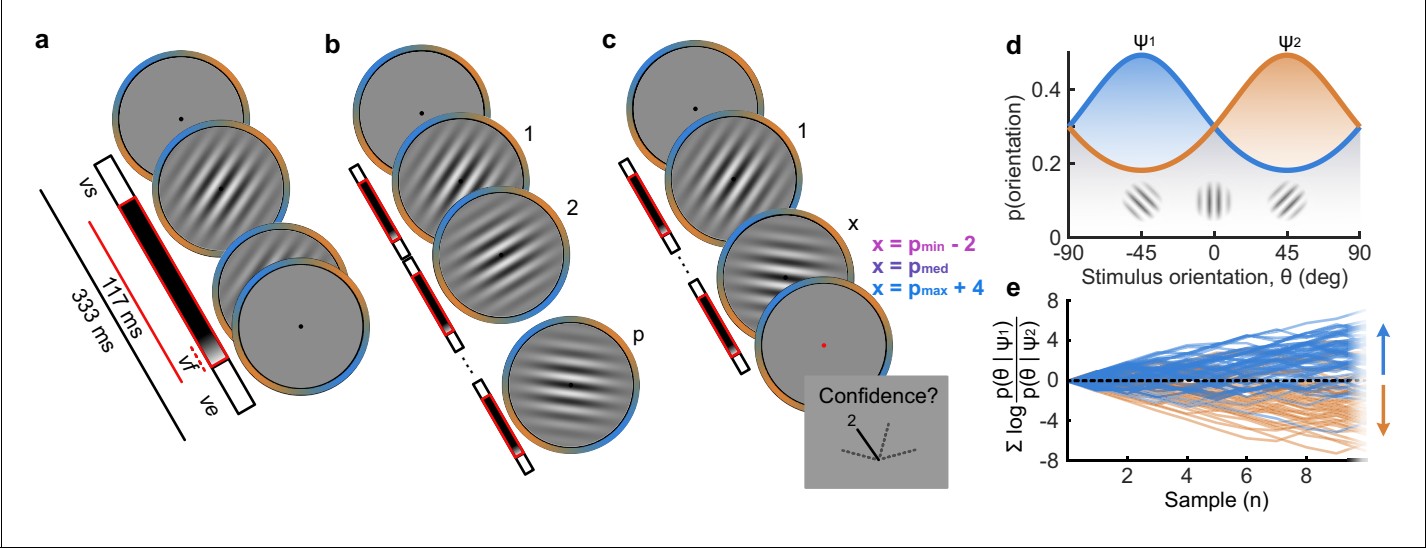

**Figure 1.** Procedure. (a) Stimulus presentation: stimuli were presented at an average rate of 3 Hz, but with variable onset and offset ($vs \in [83, 133]$ ms, $vs_s + ve_{s-1} \geq 216$ ms; see Materials and methods). Stimuli were presented within a circular annulus which acted as a colour guide for the category distributions. The colour guide and the fixation point were present throughout the trial. (b) Free task: on each trial observers were presented with a sequence of oriented Gabors, which continued until the observer entered their response (or 40 samples were shown). 100 sequences were predefined and repeated three times. (c) Replay task: The observer was presented with a specific number of samples and could only enter their response after the cue (fixation changing to red). The number of samples ($x$) was determined relative to the number the observer chose to respond to on that same sequence in the Free task ($p$). There were three intermixed conditions, Less ($x = p_{min} - 2$; where $p_{min}$ is the minimum $p$ of the three repeats), Same ($x = p_{med}$; where $p_{med}$ is the median $p$) and More ($x = p_{max} + 4$; where $p_{max}$ is the maximum $p$ of the three repeats of that predefined sequence). (d) Categories were defined by circular Gaussian distributions over the orientations, with means -45˚ ($\psi_1$, blue) and 45˚ ($\psi_2$, orange), and concentration $\kappa = 0.5$. The distributions overlapped such that an orientation of 45˚ was most likely drawn from the orange distribution but could also be drawn from the blue distribution with lower likelihood. (e) The optimal observer accumulates the difference in the evidence for each category, which is defined as the log probability of the sample orientation ($\theta$) given the distributions. The perceptual decision is determined by the sign of the accumulated evidence, where the evidence accumulated across more samples better differentiates the true categories (example evidence traces are coloured by the true category).

for evaluating their confidence. Replicating these previous results (*Balsdon et al., 2020*), we found that perceptual decision sensitivity (d') was significantly decreased with just two fewer stimuli in the Less condition compared to those same ($p_{min}$) trials in the Free task (Wilcoxon sign rank $Z = 3.88$, $p < 0.001$, Bonferroni corrected for three comparisons), but four additional stimuli in the More condition resulted in only a small but not significant increase compared to the $p_{max}$ trials in the Free task ($Z = -1.53$, $p = 0.13$, uncorrected). There was also no significant difference for the Same condition ($Z = 1.21$, $p = 0.23$, uncorrected; *Figure 2a*).

This lack of substantial increase in performance in the More condition could be the result of either a performance ceiling effect or a premature commitment to the perceptual decision. The former explanation reflects a limitation of the perceptual evidence accumulation process, whereas the latter refers to an active mechanism that ignores the final sensory evidence. We compared these two hypotheses using a computational modelling approach (*Balsdon et al., 2020*; see Materials and methods). Specifically, we compared a model in which performance in the More condition is limited by the suboptimalities evident from the Same and the Less conditions (inference noise, and temporal integration bias, see Materials and methods and Appendix 1), to a model in which performance could be impacted by a covert bound at which point observers commit to a decision irrespective of additional evidence. Cross-validated model comparison provided significant evidence that observers were implementing a covert bound (mean relative increase in model log-likelihood = 0.048, bootstrapped $p = 0.001$, *Figure 2c*). The winning model provided a good description of the data (red open markers in *Figure 2a*, and individual participants in *Figure 2e*).

In contrast to what we found for the perceptual decision, there was no evidence that observers were implementing a covert bound on confidence: Implementing the same bound as the perceptual decision did not improve the fit (relative improvement with bound = −0.007, bootstrapped p =

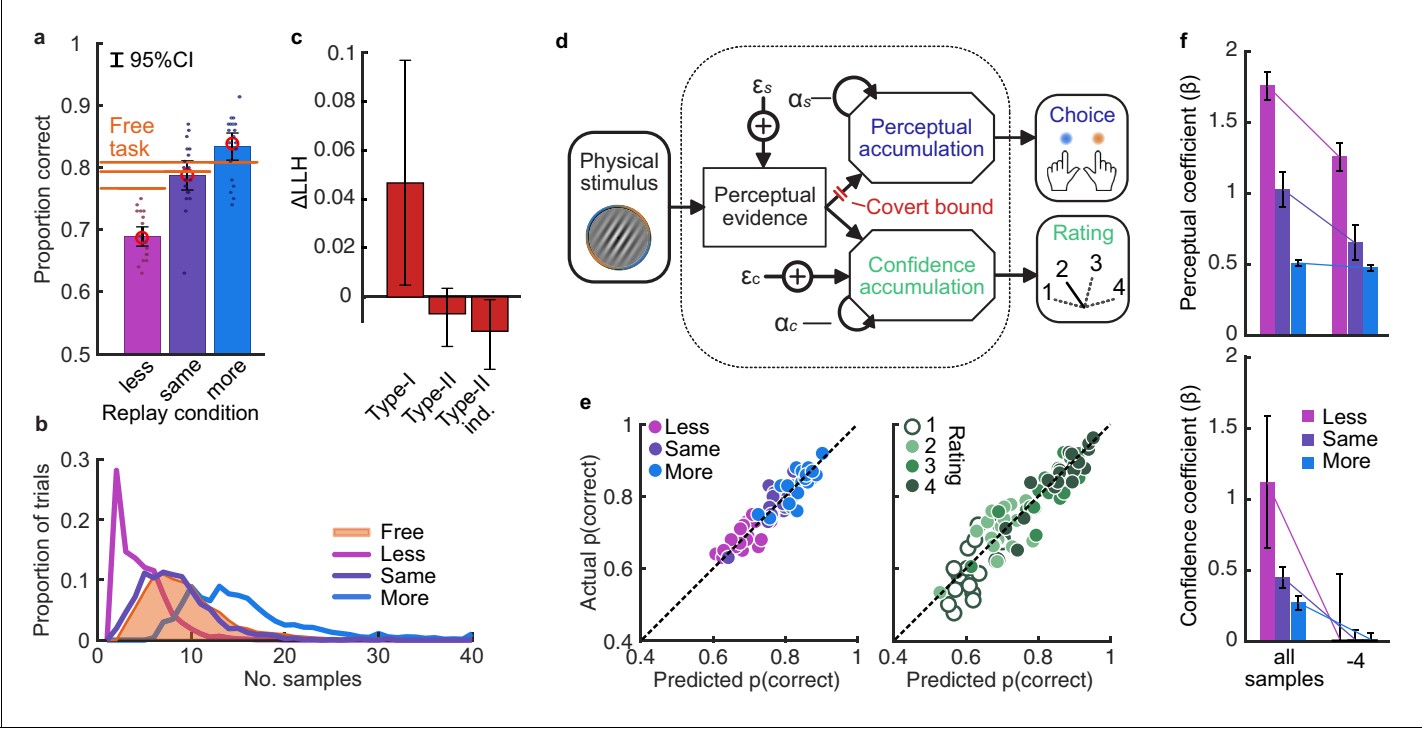

**Figure 2.** Behaviour and computational modelling. (a) Proportion correct in each condition of the Replay task, relative to the Free task (orange horizontal lines). Individual data are shown in scattered points, error bars show 95% between- (thin) and 95% within- (thick) subject confidence intervals. Open red markers show the model prediction. (b) Distributions of the number of samples per trial in the Free task, and Replay task conditions (over all observers). (c) Difference in log-likelihood of the models utilising a covert bound relative to the models with no covert bound. On the left, the model fitting perceptual decisions only. The middle bar shows the difference in log-likelihood of the fit to confidence ratings with identical perceptual and confidence bounds. The right bar shows the difference in log-likelihood of the fit to confidence ratings of the model with an independent bound for confidence evidence accumulation. Error bars show 95% between-subject confidence intervals. (d) The computational architecture of perceptual and confidence decisions, based on model comparison. Perceptual and confidence decisions accumulate the same noisy perceptual evidence, but confidence is affected by additional noise ($\varepsilon_c$) and a separate temporal bias ($\alpha_c$). This partial dissociation allows confidence evidence accumulation to continue after the observer has committed to a perceptual decision. (e) Predicted proportion correct compared to actual proportion correct for each observer, based on the fitted model parameters of the final computational model. The left panel shows proportion correct split by condition, and the right, split by confidence rating. (f) Regression coefficients from the GLM analysis showing the relationship between the optimal evidence L, and observers' perceptual (top) and confidence (bottom) responses for trials split by condition. The right set of bars show the same analysis but with evidence accumulated up to four samples from the response cue.

0.11, uncorrected) and an independent bound actually significantly *reduced* the fit compared to continued accumulation (relative improvement = −0.014, p = 0.022, Bonferroni corrected for two comparisons; *Figure 2c*). We obtained further distinctions between perceptual and confidence processes through computational modelling: additional noise was required to explain the confidence ratings, along with a separate temporal bias. The best description of both perceptual and confidence responses was provided by a partially dissociated computational architecture (full details in Appendix 1), where perceptual and confidence decisions are based on the same noisy representation of the sensory evidence, but confidence accumulation incurs additional noise and can continue after the completion of perceptual decision processes (*Figure 2d*, and the predictions of this model for individual participants are show in *Figure 2e*). These computational differences between perceptual decisions and confidence evaluations suggest deviations between the internal evidence on which observers base their perceptual and confidence decisions (see Appendix 2 for model simulations).

These modelling results are supported by an analysis using general linear models to examine the relationship between the optimal presented evidence, *L*, and observers' behaviour in the perceptual decision and confidence evaluation. As stated above, *L* is the evidence that which maximises the probability of a correct response: the accumulated difference in the log probabilities of the

presented orientations given the category distribution (*Figure 1e*). First, we find the presented evidence accumulated over all samples does explain substantial variance in observers' perceptual decisions (average $\beta$ = 0.77, $t$(19) = 6.48, p < 0.001), and confidence evaluations (with the evidence signed by the perceptual response; $\beta$ = 0.24, $t$(19) = 6.46, p < 0.001). This suggests that the internal evidence that observers were using to make their responses, L*, correlated significantly with the optimal evidence L (as has been found previously; *Drugowitsch et al., 2016*). Second, the total accumulated evidence in the More condition was not a significantly better predictor of the observers' perceptual decisions than the evidence up to four samples prior to the response (average difference in $\beta$ = 0.034, $t$(19) = 1.63, p = 0.12), while for the Same and Less conditions the total accumulated evidence was a significantly better predictor (Less: $t$(19) = 4.99, p < 0.001; Same: $t$(19) = 3.11, p = 0.006; causing a significant interaction between condition and sample accumulated to F (2,38) = 10.348, p = 0.001, Bonferroni corrected for three comparisons, *Figure 2f*, top). This supports the finding from model comparison and behaviour that observers implemented a covert bound on perceptual evidence accumulation. And finally, this interaction was not present when examining how the presented evidence affected confidence evaluations (F(2,38) = 3.124, p = 0.09, uncorrected, *Figure 2f*, bottom). Rather, the accumulated evidence up to the final sample in the More condition was a significantly better predictor of confidence than the evidence accumulated to four samples from the response (average difference in $\beta$ = 0.26, $t$(19) = 5.33, p < 0.001), supporting the prediction from the computational model analysis that observers integrated all the presented evidence for evaluating confidence.

## EEG signatures of premature perceptual decision commitment

The analysis of behaviour and computational modelling so far has suggested that observers were committing to their perceptual decisions early in the More condition and ignoring the additional evidence for their perceptual decision. We questioned the extent of this covert decision commitment, that is, whether observers were going as far as to plan their motor response before the response cue. We examined the neural signatures of the planning and execution of motor responses using a linear discriminant analysis of the spectral power of band-limited EEG oscillations (see Materials and methods). Initial analysis suggested the spectral power in the 8 to 32 Hz frequency range (the 'alpha' and 'beta' bands) could be used to classify perceptual decisions based on lateralised differences over motor cortex (Appendix 5). A classifier was trained to discriminate observers' perceptual decisions at each time-point in a four second window around the response in the Free task (3 s prior to 1 s after). This classifier was then tested across time in each condition of the Replay task, to trace the progression of perceptual decision-making in comparison to the Free task (where decisions are directly followed by response execution). If covert decisions lead to early motor response preparation, we would expect asymmetries in cross-classification performance on trials where the observer was likely to have covertly committed to a decision (in the More condition) compared to those trials in which they were unlikely to have committed to their decision (in the Less condition). Indeed, there were opposite asymmetries in the cross-classification of the Less and the More conditions (*Figure 3a*). Statistical comparison revealed substantial clusters of significant differences (*Figure 3b*): Training around −0.78 to 0.44 s from the time of the response in the Free task led to significantly better accuracy testing in the More condition than in the Less condition, prior to when the response was entered (for the cluster testing at −2.5 to −1.6 s $Z_{ave}$ = 2.04, $p_{cluster}$ = 0.002; testing at −1.5 to −1 s, $Z_{ave}$ = 1.95, $p_{cluster}$ = 0.01; testing at −0.8 to −0.3, $Z_{ave}$ = 2.32, $p_{cluster}$ < 0.001). This pattern of findings suggests that observers were not only committing to their perceptual decision early, but already preparing their motor response.

As an exploratory analysis, we took the strength of the classifier prediction trained and tested at the time of the response as a trial-wise measure of the decision variable used by the participant to enter a response. We reasoned that the amount of evidence in favour of the decision could influence the assidity with which observers enter their response. We found that the optimal evidence L, accumulated over all samples, could predict the strength of the classifier prediction at response time (mean $\beta$ = 0.11, $t$(19) = 3.89, p < 0.001; *Figure 3c*). For the Same and Less conditions, the weight on the accumulated evidence appeared to decrease as evidence was accumulated to samples further prior from the response. But, in the More condition, the evidence accumulated up to four samples prior to the response still predicted the strength of the classifier prediction ($t$(19) = 3.81, p = 0.001). This difference between conditions over samples is evidenced by a significant interaction based on a

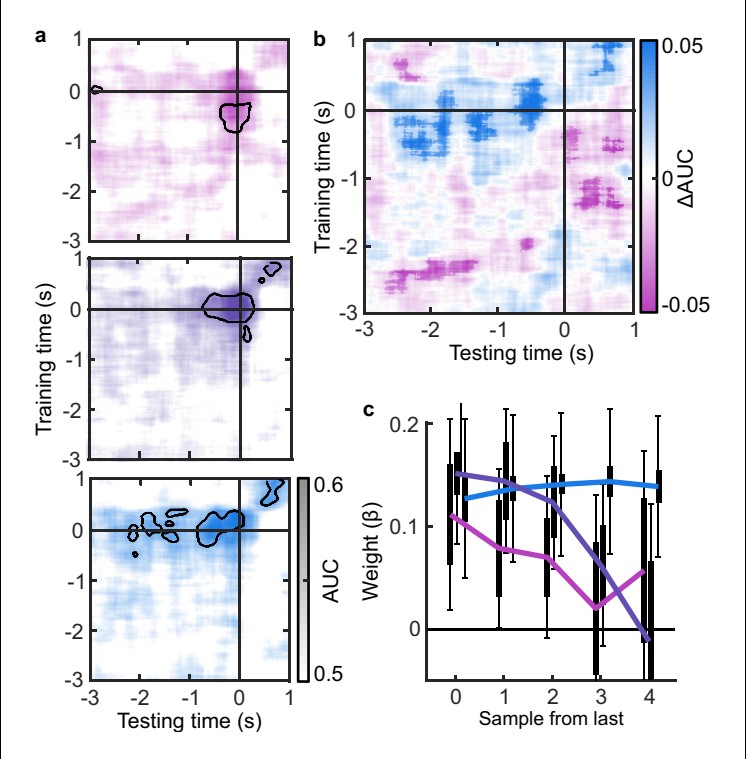

**Figure 3.** EEG signatures of premature perceptual decisions. (**a**). Classifier AUC training at each time-point in the Free task and testing across time in the Less (top), Same (middle), and More (bottom) conditions of the Replay task. Black contours encircle regions where the mean is 3.1 standard deviations from chance (0.5; 99% confidence). (**b**) Difference in AUC between the More and Less conditions. Cluster corrected significant differences are highlighted. (**c**) The relationship between the evidence accumulated up to n samples prior to the response cue and the strength of the neural signature of response execution in each condition. Error bars show 95% within- (thick) and between-subject (thin) confidence intervals.

repeated measures ANOVA ($F(8,152)$ = 2.429, p = 0.05, after Bonferroni correction for three comparisons). Leading up to the response, the accumulated evidence becomes increasingly predictive of the strength of the classifier prediction, except in the More condition, where this prediction is already accurate up to four samples prior to the response: After committing to a perceptual decision, the observer's perceptual response is no longer influenced by additional evidence.

## Representations of decision evidence in EEG signals

Our main goal was to isolate the neural signatures of the computation of confidence. Observers' behaviour varied with the optimal evidence $L$ presented to them, but the internal evidence on which they based their perceptual decisions and confidence evaluations, $L^*$, clearly deviated from $L$. In other words, the observers' behavioural performance was not optimal. To identify the neural computations underlying human behaviour, we therefore began by isolating the neural signals which correlate with $L$. We then isolated where and when deviations in the neural representation of $L$ covary with deviations in $L^*$ - the internal evidence reflected in observers' behaviour.

To perform this task the optimal observer must encode the orientation of the stimulus, estimate the decision update based on the categories, and add this to the accumulated evidence for discriminating between the categories (*Wyart et al., 2012*; *Wyart et al., 2015*). We examined the neural representation of these optimal variables using a regression analysis with the EEG signals (evoked response, bandpass filtered between 1 and 8 Hz, see Materials and methods). At each time point, we used the relationship between the pattern of neural activity and the encoding variables on 90% of the data to predict the encoding variables on the remaining 10% of the data (10-fold cross validation). The precision of the neural representation was calculated as the correlation between the

predicted encoding variable and actual encoding variable in the held-out data, across all 10 folds (see Materials and methods). *Figure 4a* shows the time course of the precision of the neural representation of stimulus orientation, momentary decision update, and accumulated evidence ($L$), locked to stimulus onset. The precision of the representations of these variables showed distinct time courses and relied on distinct patterns of EEG activity over scalp topography (*Figure 4b*). There was a transient representation of stimulus orientation localised over occipital electrodes. The representation of the momentary decision update was maintained for a longer duration, initially supported by occipital electrodes, then increasingly localised over central-parietal electrodes. The representation of the accumulated evidence was sustained even longer and relied on both frontal and occipital electrodes.

The internal evidence on which observers base their response, $L*$, can differ from the optimal evidence, $L$. When the eventual behavioural response differs from that predicted by $L$, $L*$ is likely to be more different from $L$. A neural representation of $L$ that reflects $L*$ (that is, reflecting the underlying processing responsible for behaviour) should also be less precise for samples in these trials. For each variable, we estimated the representation precision separately for epochs leading to behavioural responses that differed from the optimal response (based on $L$), and responses that matched those of the optimal observer (Replay task epochs only; *Figure 4c*; Appendix 3). For perceptual decisions, the optimal observer responds with the correct category. For confidence evaluations, the optimal observer gives high confidence on trials with greater than the median evidence (over all trials) for their perceptual response. The precision of the representation of stimulus orientation did not

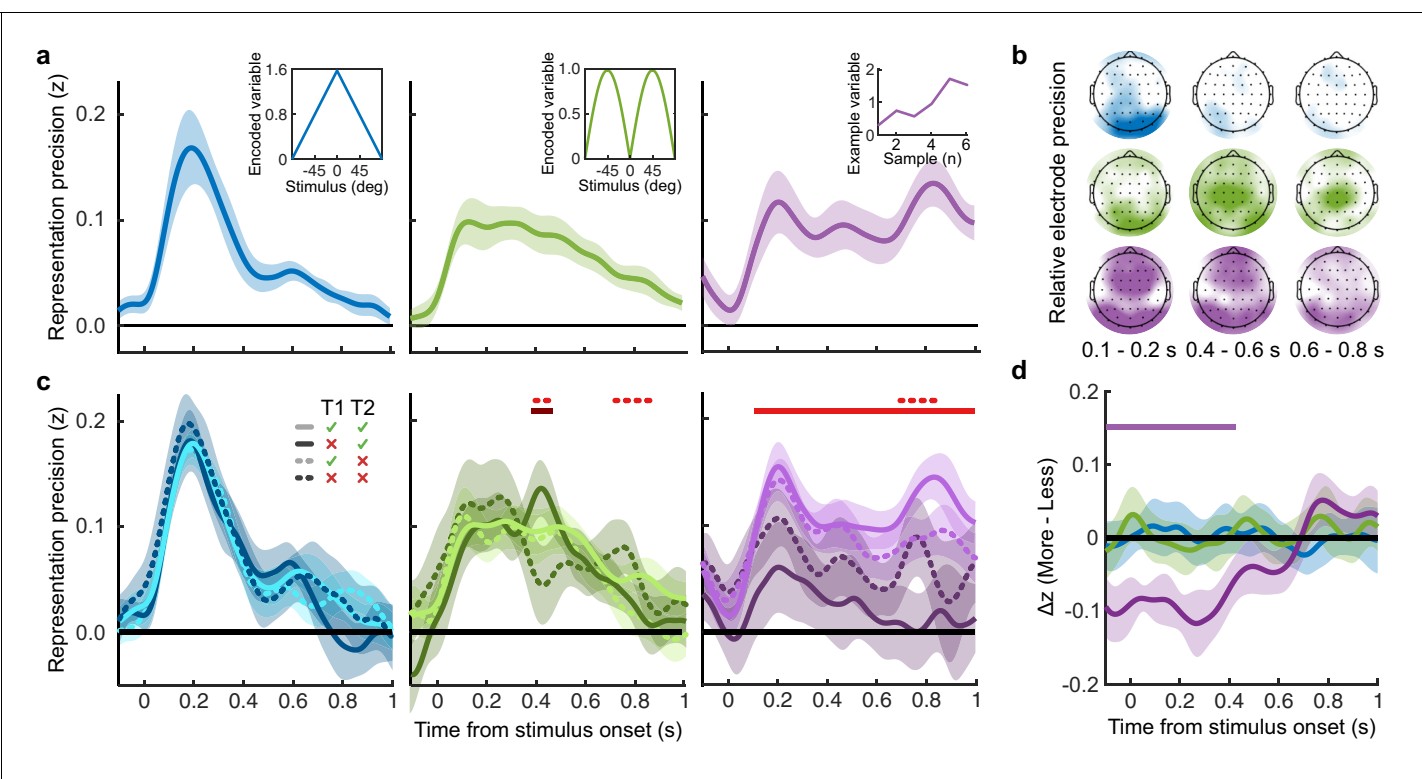

**Figure 4.** Representation of decision variables. (**a**) Representation precision (Fischer transformed correlation coefficient, $z$) of stimulus orientation (blue, left), momentary decision update (green, middle), and accumulated decision evidence (purple, right). The encoded variables are shown in the insets (the accumulated evidence is the cumulative sum of the momentary evidence signed by the response, only one example sequence is shown). Shaded regions show 95% between-subject confidence intervals. (**b**) Relative electrode representation precision over three characteristic time windows (100–200 ms, left; 400–600 ms, middle; and 600–800 ms, right). (**c**) Representation precision for epochs leading to optimal and suboptimal perceptual (T1) and confidence (T2) responses. Lighter lines show perceptual decisions that match the optimal response, dashed lines show suboptimal confidence ratings. Dashed red horizontal lines show significant interactions between perceptual and confidence suboptimality. The light red horizontal line shows the significant effect of suboptimal perception and the dark red horizontal line shows the significant effect of suboptimal confidence. Shaded regions show 95% within-subject confidence intervals. (**d**) Difference in decoding precision between the More and the Less conditions for epochs corresponding to the last four samples of the trial. The purple horizontal line shows the significant difference in decoding of accumulated evidence.

significantly vary based on whether behaviour matched the optimal response. The representation precision of the momentary decision update showed a significant effect for the perceptual decision from 380 to 468 ms ($F_{avg}(1,19)$ = 7.97, $p_{cluster}$ = 0.008) and a significant interaction between perceptual and confidence responses from 396 to 468 ms ($F_{avg}(1,19)$ = 6.66, $p_{cluster}$ = 0.022) and from 716 to 856 ms ($F_{avg}(1,19)$ = 10.75, $p_{cluster}$ < 0.001). The largest effects were seen in the representation precision of the accumulated evidence. Representation precision was significantly reduced in epochs leading to non-optimal perceptual decisions from 108 ms post stimulus onset to the end of the epoch ($F_{avg}(1,19)$ = 13.65, $p_{cluster}$ <0.001). In addition, there was a significant interaction with confidence from 696 to 836 ms ($F_{avg}(1,19)$ = 8.72, $p_{cluster}$ = 0.005). The precision of the EEG representations therefore showed distinct associations with behaviour.

The presence of a covert bound implies that, after the observer commits to a decision, they no longer incorporate additional evidence for that decision. We should therefore see significant decreases in the precision of representations that specifically contribute to perceptual evidence accumulation. Indeed, the precision of the early representation of accumulated evidence was significantly attenuated for the last four samples of the More condition (in which observers were likely to have already committed to a decision), compared to the last four samples of the Less condition (where observers were unlikely to have committed to a decision; from the start of the epoch to 424 ms, *Figure 4d*; $t_{avg}(19)$ = −5.19, $p_{cluster}$<0.001). These differences in representation precision were not present for the encoding of stimulus orientation, nor the decision update, suggesting that these processes may reflect input to perceptual evidence accumulation, but not the accumulation process itself. As a control analysis, this decreased precision was not evident in a comparison of the first four samples (Appendix 6), suggesting this effect on the representation of accumulated evidence is specific to those samples likely to have occurred after perceptual decision commitment, as opposed to those samples in More condition trials per se. Together, these comparisons suggest that different aspects of these evolving EEG representations of decision variables are related to the neural processes for perception and confidence.

## Neural processes for confidence

The analysis above shows that the EEG representation of accumulated evidence reflected greater differences from the optimal presented evidence *L* in trials where behaviour does not match the optimal response. This suggests that the corresponding neural signals reflect more closely *L** (the internal evidence actually used by observers to decide) than *L*. To isolate the neural signals which reflect *L**, we assume that *L** approximates *L* with normally distributed errors, and that these errors have larger variance on trials leading to responses that do not match the optimal evidence *L* (a similar approach as in *van Bergen et al., 2015*). We used multivariate Bayesian scan statistics (*Neill, 2011*; *Neill, 2019*) to cluster signals in space (electrode location) and time where the variance from *L* in the neural representation corresponded to deviations in *L**, based on behaviour. The statistic tested whether the variability in the neural representation was related to *L** to a greater extent than could be explained by measurement noise alone (see Appendix 7 for further details). In this way, the statistic isolates signals more closely related to *L** than can be explained by *L*, taking into account the noise affecting our measurement of these neural signals.

For perceptual decision-making, signals related to *L** were initially clustered over posterior electrodes, becoming dispersed over more anterior electrodes late in the epoch (*Figure 5a*, top). For confidence, we found two co-temporal clusters in posterior and anterior electrodes emerging from 668 ms to 824 ms from stimulus onset (*Figure 5a*, bottom). In *Figure 5a*, we highlight an early posterior cluster of signals strongly related to *L** for perceptual decisions, that was not diagnostic of confidence evaluations (in fact the evidence was in favour of the null hypothesis; summed log likelihood ratio = −1176). We obtained cluster-wide representations of *L* from the signals in this early posterior cluster and the two confidence related clusters. The precision of these representations is shown in *Figure 5b*, left. That the information from these clusters is not redundant is evident from the fact that combining the clusters improves the representation precision (*Figure 5b*). For simplicity, we combined the two confidence clusters for further analysis. Similar to the previous analysis (*Figure 4d*), the representation precision of the early posterior cluster was attenuated for the last four samples of the More condition. But, the representation precision of the confidence cluster was maintained (a repeated measures ANOVA revealed a significant interaction between cluster and condition for decoding precision in the last four samples, $F(1,19)$ = 32.00, p = 0.001, Bonferroni

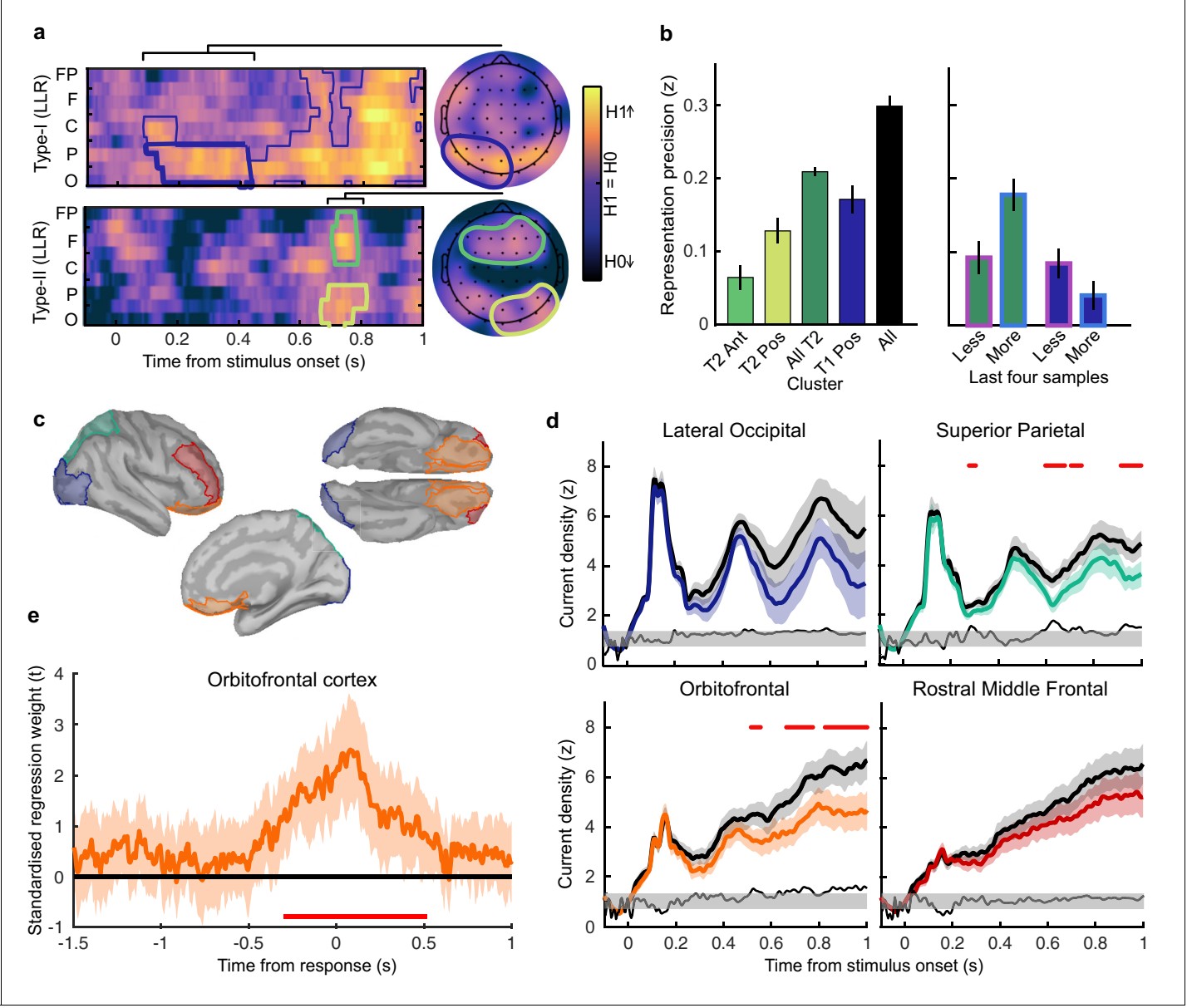

**Figure 5.** Clusters of behaviourally relevant representations and their sources. (a) Log likelihood ratio (LLR) of the data given the hypothesis that decoding precision varies with behavioural suboptimalities, against the null hypothesis that decoding precision varies only with measurement noise. Perceptual (Type-I) behaviour is shown on top and confidence (Type-II) behaviour is shown on the bottom. Clusters where the log posterior odds ratio outweighed the prior are circled, only the bold area of the perceptual cluster was further analysed. Time series (left) show the maximum LLR of electrodes laterally, with frontal polar electrodes at the top descending to occipital electrodes at the bottom. Scalp maps (right) show the summed LLR over the indicated time windows. (b) Left: representation precision (z) training and testing on signals within the clusters. Colours correspond to the circles in (a), with the dark green bar showing the combined decoding precision of the anterior and posterior confidence clusters, and the black bar showing the combined representation precision of all clusters. Right: Representation precision of the last four samples in the Less and the More conditions for the combined confidence representation and the perceptual representation. Error bars show 95% within-subject confidence intervals. (c) ROIs (defined by mindBoggle coordinates; *Klein et al., 2017*): lateral occipital cortex (blue); superior parietal cortex (green); orbitofrontal cortex (orange); and rostral middle frontal cortex (red). (d) ROI time series for Noise Max (black) and Noise Min (coloured) epochs, taking the average rectified normalised current density (z) across participants. Shaded regions show 95% within-subject confidence intervals, red horizontal lines indicate cluster corrected significant differences. Standardised within-subject differences are traced above the x-axis, with the shaded region marking z = 0 to z = 1.96 (95% confidence). (e) Standardised regression weight (t-statistic) of the GLM comparing observers' confidence ratings to those predicted from the activity localised to the orbitofrontal cortex. The shaded region shows the 95% between-subject confidence interval, and the red horizontal line marks the time-window showing cluster-corrected significant differences from 0.

corrected for three comparisons). These results are consistent with dissociable stages of neural processing for confidence evaluation and perceptual decision-making, and support the computational modelling in suggesting a partial dissociation between the internal evidence used for making perceptual decisions and confidence evaluations.

We used the representation from the confidence cluster as an estimate of the internal evidence on which observers base their confidence ratings. We then took the difference from $L$ in the estimate of $L*$ from the cluster representation as an estimate of the single-sample inference error. This estimate of the single-sample inference error was significantly correlated with the single-sample inference error estimated from the computational model of confidence ratings ($t(19) = 5.12$, $p < 0.001$), and this correlation was significantly greater than the correlation with the error estimated from the model of perceptual decisions alone ($t(19) = 2.62$, $p = 0.017$; see Appendix 8). This suggests that this cluster representation is indeed reflecting activity specific to the computation of confidence.

We asked what processes were responsible for driving variability in the internal evidence for confidence beyond what could be explained by the evidence presented to the observer. We selected 'Noise Min' and 'Noise Max' epochs as the top and bottom quartile of epochs sorted by the estimate of the inference error from the cluster representation, and examined the source-localised EEG activity across these epochs. The presented sensory evidence was similar across Noise Min and Noise Max epochs (see Appendix 8), but the additional variability in the Noise Max epochs pushes the represented evidence further from the mean, and should therefore correspond to a greater absolute normalised signal. We estimated the sources of activity in the Noise Min and Noise Max epochs using a template brain (see Materials and methods) and tested for differences in the rectified normalised current density in ROIs defined based on the previous literature (*Figure 5c*; *Graziano et al., 2015*; *Gherman and Philiastides, 2018*; *Herding et al., 2019*, see Appendix 9). As expected, Noise Max epochs showed a greater increase in current density power over time. Significant differences first emerged in the superior parietal cortex (*Figure 5d*; 276 to 304 ms; $t_{avg}(19) = 2.37$, $p_{cluster} = 0.016$, re-emerging at 596 to 748 ms; $t_{avg}(19) = 2.53$, $p_{cluster} = 0.016$; and 912 ms; $t_{avg}(19) = 2.50$, $p_{cluster} = 0.014$), and then in the orbitofrontal cortex (OFC; 516 to 556 ms; $t_{avg}(19) = 2.30$, $p_{cluster} = 0.022$, re-emerging at 660 to 772 ms; $t_{avg}(19) = 2.79$, $p_{cluster} = 0.032$, and 824 to 1000 ms; $t_{avg}(19) = 2.60$, $p_{cluster} = 0.022$). No differences in the rostral middle frontal cortex nor lateral occipital cortex survived cluster correction.

Whilst the activity localised to the superior parietal cortex reflected stimulus driven computations (the consecutive peaks correspond temporally to the response to subsequent stimuli), the activity localised to the orbitofrontal cortex was more indicative of an accumulation process across samples (a smoother increase in signal over time). As an exploratory analysis, we tested whether the activity localised to the orbitofrontal cortex could predict observers' confidence ratings, presumably by accumulating evidence for evaluating confidence up to the observers' perceptual decision response. Indeed, the activity localised to the orbitofrontal cortex predicted observers' confidence ratings, based on the predictions of a generalised linear model with 90/10 cross validation: the standardised regression coefficients increased up to and continued after the perceptual decision response (*Figure 5e*, a significant cluster was located from −300 to 520 ms around the time of the response; $t_{ave}(19) = 3.46$, cluster-corrected $p < 0.001$).

## Discussion

We examined the dynamic neural signals associated with the accumulation of evidence for evaluating confidence in perceptual decisions. Observers were required to integrate evidence over multiple samples provided by a sequence of visual stimuli. When observers were unable to control the amount of evidence they were exposed to, they employed a covert decision bound, committing to perceptual decisions when they had enough evidence, even if stimulus presentation continued. We had previously shown evidence for this premature decision commitment based on behaviour and computational modelling (*Balsdon et al., 2020*). We replicated these results here, and further examined the neural signatures of covert decision making. We found that the distribution of spectral power associated with the preparation and execution of motor responses in the Free task (where the response is entered as soon as the decision is made) could be used to accurately predict responses in the More condition of the Replay task over 1 s prior to when the response was entered, and with significantly greater sensitivity than in the Less condition (when observers were unlikely to have

committed to a decision early). This suggests that covert decisions could trigger the motor preparation for pressing the response key. Moreover, the strength of the eventual motor response signal could be predicted by earlier decision evidence in the More condition, as if observers are maintaining some representation of the decision evidence whilst waiting to press the response key.

Based on the evoked representation of accumulated evidence, perceptual decision accuracy relied on a flow of information processing from early occipital and parietal signals, which then spread through to anterior electrodes. When observers committed to perceptual decisions prematurely, only the early part of the representation of accumulated evidence was attenuated. This selective dampening of the representation of accumulated evidence following premature decision commitment delineates which computations are devoted solely to the perceptual decision process, and which computations reflect the input to the decision process: The representations of stimulus orientation and decision update (*Wyart et al., 2012*; *Wyart et al., 2015*; *Weiss et al., 2021*), which are necessary input for the perceptual decision, did not substantially change after committing to a perceptual decision. This initial perceptual processing stage likely remained important for the continued accumulation of evidence for evaluating confidence (even after the completion of perceptual decision processes), though it could also be that these processes are automatically triggered by stimulus onset irrespective of whether the evidence is being accumulated for decision-making.

Confidence should increase with increasing evidence for the perceptual decision. It is therefore unsurprising that the neural correlates of confidence magnitude have found similar EEG markers as those related to the accumulation of the underlying perceptual decision evidence: the P300 (*Gherman and Philiastides, 2015*; *Desender et al., 2016*; *Desender et al., 2019*; *Zakrzewski et al., 2019*; *Rausch et al., 2020*) and Central Parietal Positivity (CPP; *Boldt et al., 2019*; *Herding et al., 2019*, indeed we show a similar effect in Appendix 4). The analysis presented in this manuscript targeted confidence precision rather than confidence magnitude, by assessing confidence relative to an optimal observer who gives high confidence ratings on trials where the evidence in favour of the perceptual choice is greater than the median across trials. We isolated part of the neural representation of accumulated evidence where imprecision relative to the optimal presented evidence predicted greater deviations from optimal in the internal representation of evidence used for confidence evaluation implied from behaviour. The internal evidence predicted from this neural representation was also more strongly related to the evidence for confidence than the evidence used for perceptual decisions based on the computational model fit to describe behaviour.

We analysed the sources of activity more closely representing the internal evidence on which the confidence evaluation was based than the optimal presented evidence. Activity localised to the superior parietal and orbitofrontal cortices was found to track this internal evidence for confidence throughout decision-making. This is not at odds with the previous literature: The difference in superior parietal cortex could be linked with findings from electrophysiology that suggest that confidence is based on information coded in parietal cortex, where the underlying perceptual decision evidence is integrated (*Kiani and Shadlen, 2009*; *Rutishauser et al., 2018*; though at least a subset of these neurons reflect bounded accumulation, which is in contrast with the continued confidence accumulation described in this experiment; *Kiani et al., 2008*). Early electrophysiological investigation into the function of the orbitofrontal cortex revealed neural coding associated with stimulus value (*Thorpe et al., 1983*), which has since been linked with a confidence-modulated signal of outcome-expectation (*Kepecs et al., 2008*; and in human fMRI; *Rolls et al., 2010*) and recently, shown to be domain-general (single OFC neurons were associated with confidence in both olfactory and auditory tasks; *Masset et al., 2020*). The source localisation analysis therefore connects previous findings, indicating confidence feeds off an evidence accumulation process, culminating in higher order brain areas that appear to function for guiding outcome-driven behaviour based on decision certainty.

These neural signatures of confidence evidence encoding were present throughout the process of making a perceptual decision. This is in line with more recent evidence suggesting that confidence could be computed online, alongside perceptual evidence accumulation (*Zizlsperger et al., 2014*; *Gherman and Philiastides, 2015*; *Balsdon et al., 2020*), as opposed to assessing the evidence in favour of the perceptual decision only after committing to that decision. Computational model comparison supported this interpretation, showing the best description of confidence behaviour was an accumulation process that was partially dissociable from perceptual evidence accumulation (Appendix 1; replicating our previous analysis, *Balsdon et al., 2020*). This partial dissociation mediates the ongoing debate between single-channel (for example, *Maniscalco and Lau, 2016*) and dual-channel

(for example, *Charles et al., 2014*) models, as it constrains confidence by perceptual suboptimalities, at the same time as allowing additional processing to independently shape confidence. The combination of this partial dissociation and online monitoring could allow for metacognitive control of perceptual evidence accumulation, to flexibly balance perceptual accuracy against temporal efficiency, by bounding perceptual evidence accumulation according to contemporaneous confidence.

Using this protocol, we were able to delineate two distinct representations of accumulated evidence which correspond to perceptual decision-making and confidence evaluations. These neural representations were partially dissociable in that the perceptual representation neglected additional evidence following premature decision commitment whilst the confidence representation continued to track the updated evidence independently of decision commitment. This partial dissociation validates the predictions of the computational model and provides a framework for the cognitive architecture underlying the distinction between perception and confidence. That the neural resources involved in the confidence representation can be recruited independently of perceptual processes implies a specific neural circuit for the computation of confidence, a necessary feature of a general metacognitive mechanism flexibly employed to monitor the validity of any cognitive process.

## Materials and methods

### Participants

A total of 20 participants were recruited from the local cognitive science mailing list (RISC) and by word of mouth. No participant met the pre-registered (https://osf.io/346pe/) exclusion criteria of chance-level performance or excessive EEG noise. Written informed consent was provided prior to commencing the experiment. Participants were required to have normal or corrected to normal vision. Ethical approval was granted by the INSERM ethics committee (ID RCB: 2017-A01778-45 Protocol C15-98).

### Materials

Stimuli were presented on a 24' BenQ LCD monitor running at 60 Hz with resolution 1920 x 1080 pixels and mean luminance 45 cd/m². Stimulus generation and presentation was controlled by MATLAB (Mathworks) and the Psychophysics toolbox (*Brainard, 1997*; *Pelli, 1997*; *Kleiner et al., 2007*), run on a Dell Precision M4800 Laptop. Observers viewed the monitor from a distance of 57 cm, with their head supported by a chin rest. EEG data were collected using a 64-electrode BioSemi Active-Two system, run on a dedicated mac laptop (Apple Inc), with a sample rate of 512 Hz. Data were recorded within a shielded room.

### Stimuli

Stimuli were oriented Gabor patches displayed at 70% contrast, subtending four dva and with spatial frequency two cyc/deg. On each trial a sequence of stimuli was presented, at an average rate of 3 Hz, with the stimulus presented at full 70% contrast for a variable duration between 50 and 83 ms, with a sudden onset, followed by an offset ramp over two flips, where the stimulus contrast decreased by 50% and 75% before complete offset. Stimulus onset timing was jittered within the stimulus presentation interval such that the timing of stimulus onset was irregular but with at least 216 ms between stimuli. These timings and stimulus examples are shown in *Figure 1a*.

On each trial the orientations of the presented Gabors were drawn from one of two circular Gaussian (Von Mises) distributions centred on +/- 45˚ from vertical (henceforth referred to as the 'orange' and 'blue' distributions, respectively), with concentration κ = 0.5 (shown in *Figure 1d*). Stimuli were displayed within an annular 'colour-guide' where the colour of the annulus corresponds to the probability of the orientation under each distribution, using the red and blue RGB channels to represent the probabilities of each orientation under each distribution. Stimuli were presented in the centre of the screen, with a black central fixation point to guide observers' gaze.

### Procedure

The task was a modified version of the weather prediction task (*Knowlton et al., 1996*; *Drugowitsch et al., 2016*). Throughout the experiment, the observer's perceptual task was to categorise which distribution the stimulus orientations were sampled from. They were instructed to press

the 'd' key with their left hand (of a standard query keyboard) for the blue distribution and the 'k' key with their right hand for the orange distribution. There were two variants of the task: The Free task and the Replay task. The trials were composed of three repetitions of 100 predefined sequences of up to 40 samples (50 trials from each distribution) for each observer (300 trials per task).

In the 'Free' task, observers were continually shown samples (up to 40) until they entered their response. They were instructed to enter their response as soon as they 'feel ready' to make a decision, with emphasis on both accuracy (they should make their decision when they feel they have a good chance of being correct) and on time (they shouldn't take too long to complete each trial). A graphical description of this task is shown in *Figure 1b*.

After completing the Free task, observers then completed the Replay task. In this task they were shown a specific number of samples and could only enter their response after the sequence finished, signalled by the fixation point turning red. The number of samples was determined based on the number observers chose to respond to in the Free task. There were three intermixed conditions: In the Less condition observers were shown two fewer samples than the minimum they had chosen to respond to on that predefined sequence in the Free task; In the Same condition observers were shown the median number of samples from that predefined sequence; in the More condition observers were shown four additional samples compared to the maximum number they chose to respond to on that sequence in the Free task. After entering their perceptual response, observers were cued to give a confidence rating. The confidence rating was given on a four-point scale where 1 represents very low confidence that the perceptual decision was correct, and 4, certainty that the perceptual decision was correct. The rating was entered by pressing the 'space bar' when a presented dial reached the desired rating. The dial was composed of a black line which was rotated clockwise to each of 4 equidistant angles (marked 1–4) around a half circle, at a rate of 1.33 Hz. The dial started at a random confidence level on each trial and continued updating until a rating was chosen. A graphical description of this task is shown in *Figure 1c*.

Prior to commencing the experimental trials, participants were given the opportunity to practice the experiment and ask questions. They first performed 20 trials of a fixed number of samples with only the perceptual decision, with feedback after each response as to the true category. They then practiced the Replay task with the confidence rating (and an arbitrary number of samples). Finally, they practiced the Free task, before commencing the experiment with the Free task.

## Analysis

### Behaviour

Perceptual decisions were evaluated relative to the category the orientations were actually drawn from. Performance is presented as proportion correct, whilst statistical analyses were performed on sensitivity (d'). Sensitivity was calculated based on the proportion of hits (responding 'Category A' when category A was presented) and false alarms (responding 'Category A' when category B was presented). Confidence was evaluated relative to an optimal observer who gives high confidence when the log-likelihood of the chosen category, based on the presented orientations, is above the median across trials, and low confidence on trials with less than the median log-likelihood. More broadly, confidence should increase with increasing evidence in favour of the perceptual decision, see Appendix 3. A General Linear Model was used to validate the influence of the optimal presented evidence on perceptual decisions and confidence evaluations. The accumulated evidence up to the final sample and four samples before the response was used as a regressor for the perceptual decision assuming a binomial distribution with a probit link function. A comparable analysis was performed for confidence by binarizing confidence ratings into Low (ratings of 1 or 2) and High (ratings of 3 or 4) and taking the evidence signed by the perceptual decision.

### Computational modelling

Computational modelling followed the same procedure as *Balsdon et al., 2020*. The model parametrically describes suboptimalities relative to the Bayesian optimal observer. The Bayesian optimal observer knows the category means, $\mu_1 = -\frac{\pi}{4}, \mu_2 = \frac{\pi}{4}$, and the concentration, $\kappa = 0.5$, and takes the probability of the orientation $\theta_n$ (at sample *n*) given each category $\psi$ ($\psi = 1$ or $\psi = 2$)

$$p(\theta_n|\psi) = \frac{e^{\kappa\cos(2(\theta_n - \mu_\psi))}}{\pi I_0(\kappa)} \tag{1}$$

where $I_0(\cdot)$ is the modified Bessel function of order 0. The optimal observer then chooses the category $\psi$ with the greatest posterior probability over all samples for that trial, $T$ ($T$ varies from trial to trial). Given a uniform category prior, $p(\psi) \propto \frac{1}{2}$, and perfect anticorrelation in $p(\theta_n|\psi)$ over the categories, the log posterior is proportional to the sum of the difference in the log-likelihood for each category ($\ell_n = \ell_{n,1} - \ell_{n,2}$)

$$L = \sum_{n=1}^{T} \ell_n \tag{2}$$

where:

$$\ell_{n,\psi} = \log p(\theta_n|\psi) = \kappa\cos(2(\theta_n - \mu_\psi)) + const. \tag{3}$$

Such that the Bayesian optimal decision is 1 if $L>0$ and 2 if $L \leq 0$.

The suboptimal observer suffers inaccuracies in the representation of each evidence sample, captured by additive independent identically distributed (i.i.d) noise, $\varepsilon_n$. The noise is Gaussian distributed with zero mean, and the degree of variability parameterised by $\sigma$, the standard deviation

$$\varepsilon_n \sim N(0, \sigma^2) \tag{4}$$

The evidence over samples is also imperfectly accumulated, incurring primacy or recency biases parameterised by $\alpha$, the weight on the current accumulated evidence compared to the new sample ($\alpha>1$ creates a primacy effect). By the end of the trial, the weight on each sample $n$ is equal to

$$v_n = \alpha^{T-n} \tag{5}$$

where $T$ is the eventual total samples on that trial and $n \in [1, T]$.

In the Free task, the observer responds when accumulated evidence reaches a bound, $\Lambda$. The optimal observer sets a constant bound on proportion correct over sequence length, which is an exponential function on the average evidence over the samples accumulated. The human observer can set the scale, $b$, and the rate of decline, $\lambda$, of the bound suboptimally, resulting in

$$\Lambda_{n+} = n \times \left(a + be^{-\frac{n}{\lambda}}\right) \tag{6}$$

for the positive decision bound (the negative bound, $\Lambda_{n-} = -\Lambda_{n+}$). The likelihood $f(n)$ of responding at sample n was estimated by computing the frequencies, over 1000 samples from $\varepsilon_n$ (Monte Carlo simulation), of first times where the following inequality is verified

$$\left|\sum_{n=1}^{N}(l_n + \varepsilon_n)\cdot v_n\right| > \Lambda_n \tag{7}$$

The response time, relative to reaching the decision bound, is delayed by a non-decision time for executing the motor response, which is described by a Gaussian distribution of mean, $\mu_U$, and variance, $\sigma_U^2$.

## Model fitting

Parameters were optimised to minimise the negative log-likelihood of the observer making response $r$ on sample $n$ on each trial for each participant using Bayesian Adaptive Direct Search (*Acerbi and Ma, 2017*). The log-likelihoods were estimated using Monte Carlo Simulation, with the sensitivity of this approach being addressed in previous work (*Balsdon et al., 2020*). The full model was simplified using a knock-out procedure based on Bayesian Model Selection (*Rigoux et al., 2014*) to fix the bias (exceedance probability = 0.93) and lapse (exceedance probability >0.99) parameters (not described above, see Appendix 1).

In the Replay task, confidence ratings were fit using the same model described above, but with additional criteria determining confidence ratings, described by three bounds on the confidence

evidence, parameterised in the same manner as the decision bound. These models were then used to simulate the internal evidence of each observer from sample to sample, and the error compared to the optimal evidence (uncorrupted by suboptimalities, see Appendix 2).

## EEG pre-processing

EEG data were pre-processed using the PREP processing pipeline (*Bigdely-Shamlo et al., 2015*), implemented in EEGlab (v2019.0, *Delorme and Makeig, 2004*) in MATLAB (R2019a, Mathworks). This includes line noise removal (notch filter at 50 Hz and harmonics) and re-referencing (robust average re-reference on data detrended at 1 Hz). The data were then filtered to frequencies between 0.5 and 80 Hz, and down-sampled to 256 Hz. Large epochs were taken locked to each stimulus (−500 to 1500 ms) and each response (−5000 to 1500 ms). Independent Components Analysis was used to remove artefacts caused by blinks and excessive muscle movement identified using labels with a probability greater than 0.35 from the ICLabel project classifier (Swartz Centre for Computational Neuroscience).

## Response classification analysis

The power spectrum across frequency tapers from 1 to 64 Hz with 25% spectral smoothing was resolved using wavelet convolution implemented in FieldTrip (*Oostenveld et al., 2011*). The epochs were then clipped at −3 to 1 s around the time of entering the perceptual response. Linear discriminant analysis was performed to classify perceptual responses, using 10-fold cross validation, separately on each taper at each time-point. An analysis of the frequencies contributing to accurate classification at the time of the response revealed significant contributions from 8 to 26 Hz (Appendix 4). We therefore continued by using the power averaged across these frequency bands to train and test the classifier. Classifier accuracy was assessed using the area under the receiver operating characteristic curve (AUC). At the single-trial level, the probability of the response based on the classifier was computed from the relative normalised Euclidean distance of the trial features from the response category means in classifier decision space.

## Encoding variable regression

We used a linear regression analysis to examine the EEG correlates of different aspects of the decision evidence (encoding variables) in epochs locked to stimulus onset. Regularised ridge regression (ridge $\lambda = 1$) was used to predict the encoding variables based on EEG data, over 10-fold cross validation. The precision of the representation of each encoding variable was computed within each observer by taking the Fisher transform of the correlation coefficient (Pearson's r) between the encoded variable and predicted variable. To maximise representation precision, the data were bandpass filtered (1 – 8 Hz) and decomposed into real and imaginary parts using a Hilbert Transform (Appendix 5). For each time point, the data from all electrodes were used to predict the encoded variable. The temporal generalisation of decoding weights was examined by training at one time point and testing at another. The contribution of information from signals at each electrode was examined by training and testing on the signals at each electrode at each time point (further details in Appendix 5).

Behaviourally relevant signals were isolated by comparing representation precision at each time point and electrode for epochs leading to optimal perceptual and confidence responses, compared to responses that did not match the optimal observer. Cluster modelling was used to isolate contiguous signals where the log posterior odds were in favour of the alternative hypothesis that the representation systematically deviated further from the optimal presented evidence than what could be explained by measurement noise alone (Appendix 6). New regression weights were then calculated on signals from the entire cluster and representation errors calculated as the difference of the predicted variable from the expected value given the representation.

## Source localisation

Identifying the clusters of signals associated with confidence processes offers relatively poor spatial and temporal (given the bandpass filter; *de Cheveigné and Nelken, 2019*) resolution for identifying the source of confidence computations. Source localisation was therefore performed, using Brainstorm (*Tadel et al., 2011*). The forward model was computed using OpenMEEG (*Gramfort et al.,*

*2010*; *Kybic et al., 2005*) and the ICBM152 anatomy (*Fonov et al., 2011*; *Fonov et al., 2009*). Two conditions were compared, Noise Min and Noise Max, which corresponded to quartiles of epochs sorted by representation error in the confidence clusters (see Appendix 7 for more details). Cortical current source density was estimated from the average epochs using orientation-constrained minimum norm imaging (*Baillet et al., 2001*). ROIs in the lateral occipital, superior parietal, rostral middle frontal (including dlPFC), medial orbitofrontal, and rostral anterior cingulate cortex, were defined using MindBoggle (*Klein et al., 2017*). Statistical comparisons were performed on the bilateral ROI time series (using cluster correction and a minimum duration of 20 ms), computed over separate conditions on rectified normalised subject averages (low-pass filtered at 40 Hz).

To predict confidence magnitude from the activity localised to the orbitofrontal cortex, we recovered to current density from 20 subregions (approximately equal parcellations) of the orbitofrontal cortex in epochs locked to the time of the response. A general linear model (assuming a normal distribution with identity link) was used to predict the observers' confidence ratings on held-out data (90/10 cross-fold) from the neural activity at each time-point leading to the response. The prediction was quantified as the standardised regression weight from a new general linear model comparing the predicted and actual confidence ratings across all folds.

# Additional information

## Competing interests

Valentin Wyart: Reviewing editor, *eLife*. The other authors declare that no competing interests exist.

## Funding

| Funder | Grant reference number | Author |
|---|---|---|
| Labex | ANR-10-LABX-0087 IEC | Pascal Mamassian<br>Valentin Wyart |
| Inserm | U960 | Valentin Wyart |
| Centre National de la Recherche Scientifique | UMR 8248 | Pascal Mamassian |
| ANR | ANR-18-CE28-0015 | Pascal Mamassian |

The funders had no role in study design, data collection and interpretation, or the decision to submit the work for publication.

## Author contributions

Tarryn Balsdon, Conceptualization, Data curation, Software, Formal analysis, Investigation, Visualization, Methodology, Writing - original draft, Writing - review and editing; Pascal Mamassian, Conceptualization, Resources, Data curation, Supervision, Funding acquisition, Investigation, Methodology, Project administration, Writing - review and editing; Valentin Wyart, Conceptualization, Resources, Data curation, Software, Supervision, Funding acquisition, Investigation, Methodology, Project administration, Writing - review and editing

## Author ORCIDs

Tarryn Balsdon https://orcid.org/0000-0002-3122-6630
Pascal Mamassian https://orcid.org/0000-0002-1605-4607
Valentin Wyart http://orcid.org/0000-0001-6522-7837

## Ethics

Human subjects: Participants provided written informed consent prior to commencing the experiment. Ethical approval was granted by the INSERM ethics committee (ID RCB: 2017-A01778-45 Protocol C15-98).

Decision letter and Author response
Decision letter https://doi.org/10.7554/eLife.68491.sa1
Author response https://doi.org/10.7554/eLife.68491.sa2

## Additional files
### Supplementary files
• Transparent reporting form

### Data availability
Data is available on the Open Science Framework.

The following dataset was generated:

| Author(s) | Year | Dataset title | Dataset URL | Database and Identifier |
|---|---|---|---|---|
| Balsdon T, Mamassian P, Wyart V | 2021 | Raw data | https://doi.org/10.17605/OSF.IO/XC6G2 | Open Science Framework, 10.17605/OSF.IO/XC6G2 |

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

## Appendix 1

### Computational Model fitting

The computational model is described in full in the Materials and methods section. Briefly, the model is based on the Bayesian optimal observer with full knowledge of the category distributions (means $\mu_1$ and $\mu_2$, concentration $\kappa$), and takes as evidence the difference in the log posterior probability ($\ell_n$) of each category given the orientation ($\theta_n$)

$$\begin{aligned} \ell_n &= \ell_{n,1} - \ell_{n,2} = \kappa\cos(2(\theta_n - \mu_2)) \\ &= 2\kappa\sin(\mu_1 - \mu_2)\sin(2\theta_n - \mu_1 - \mu_2) = \sin(2\theta_n) \end{aligned} \tag{A1}$$

where chosen values ($\kappa = 0.5$, $\mu_1 = -\pi/4$, and $\mu_2 = \pi/4$) have been implemented in the last equation. Whilst the optimal observer perfectly sums the evidence over each sample, the suboptimal human observer accumulates evidence with some temporal integration bias, $\alpha$ (where $\alpha > 1$ creates a primacy effect, and $\alpha < 1$, a recency effect), and incurs inference error (noise in the estimate of the true evidence) parameterised by $\sigma$, the standard deviation of the Gaussian distribution from which each sample of noise, $\varepsilon_n$, is drawn from. The human observer may also experience some response bias, $c$ (the tendency to choose one category irrespective of the evidence), and incur lapses (pressing a random key), described by the lapse rate, $l$. The accumulated evidence, $L^*$, up to sample $n$, is suboptimally accumulated by

$$L_n^* = \alpha L_{n-1}^* + \ell_n + \varepsilon_n \tag{EA2}$$

The observer then chooses category one if $L^* > c$, except on a proportion of trials, $l$, where the response is randomly selected.

These four parameters were used to capture the differences in the human observers' responses (category choice and confidence rating) from the optimal observer who perfectly integrates all evidence presented.

In the Free task, the model was designed not only to describe the category choice, but at which sample the human observer chose to respond. This was achieved via a decision boundary, the nature of which has been addressed in previous work (*Balsdon et al., 2020*). The boundaries, $\Lambda_{n+}$ and $\Lambda_{n-}$, follow an exponential function on the average evidence over samples (which is a constant bound on the probability of a correct response), described by three parameters: the minimum, $a$, the scale, $b$, and the rate of decline, $\lambda$

$$\Lambda_{n+} = n \times \left(a + be^{-\frac{n}{\lambda}}\right) \tag{A3}$$

There is an optimal combination of these parameters to achieve any particular proportion correct across the experiment, but the human observer may set their bound suboptimally. In addition, non-decision time (the time from the last sample integrated to pressing the response key) was described by a Normal distribution with mean $\mu_U$, and variance $\sigma_U^2$. Giving an additional five parameters for describing when the observer enters their response.

We followed the same procedure as in *Balsdon et al., 2020*, involving four stages:

1. Reduce the number of free parameters with a knock-out procedure.
2. Compare (covert) Bound and No-bound models of the perceptual decision in the Replay task.
3. Identify any systematic differences in the parameters required to describe the confidence ratings, compared to the perceptual decision, in order to discern the relationship between processes for perceptual decisions and confidence.
4. Apply the same Bound vs. No-bound comparison for describing the confidence ratings.

The average parameter values and fit metrics for Stage 1. are shown in *Appendix 1—table 1*. According to this analysis, the bias ($c$) and lapse rate ($l$) were fixed. There was some evidence the boundary minimum ($a$) could be fixed in the Replay task, but the preference in the Free task was to leave it free to vary.

**Appendix 1—table 1.** Average parameter values.
Table shows the average values and the sum of BIC across participants. The large difference in the average log-likelihood (LLH) across tasks is due to the fact the Free task model was fit to both when

and what observers responded, whereas in the Replay task only the response was fit. Red values show the fixed parameters.

**Free task**

| Model | $\sigma$ | $\alpha$ | $c$ | $\mu_U$ | $\sigma_U^2$ | $a$ | $b$ | $\lambda$ | $l$ | LLH | $\sum$BIC |
|---|---|---|---|---|---|---|---|---|---|---|---|
| Full | 0.83 | 0.98 | −0.04 | 425 | 0.52 | 0.10 | 6.04 | 1.93 | 0.016 | −734.91 | 30423.01 |
| $\alpha = 1$ | 0.83 | 1.00 | 0.00 | 430 | 0.50 | 0.13 | 6.61 | 2.03 | 0.014 | −734.97 | 30311.59 |
| $c = 0$ | 0.80 | 0.92 | 0.00 | 452 | 0.54 | 0.11 | 5.28 | 2.01 | 0.017 | −736.86 | 30387.02 |
| $\mu_U = 400$ | 0.76 | 0.94 | 0.00 | 400 | 0.52 | 0.09 | 5.52 | 2.23 | 0.016 | −739.77 | 30503.40 |
| $\sigma_U^2 = 1$ | 0.69 | 0.96 | −0.02 | 435 | 1.00 | 0.10 | 6.34 | 1.97 | 0.015 | −754.18 | 31079.84 |
| $a = 0.1$ | 0.77 | 0.92 | 0.03 | 417 | 0.52 | 0.10 | 5.78 | 2.20 | 0.016 | −735.48 | 30331.75 |
| $b = 5.5$ | 0.78 | 0.94 | 0.02 | 410 | 0.64 | 0.13 | 5.50 | 1.79 | 0.013 | −742.18 | 30599.67 |
| $l = 0.001$ | 0.82 | 0.98 | 0.01 | 400 | 0.48 | 0.10 | 4.77 | 2.22 | 0.001 | −730.66 | 30139.17 |
| $c = 0$; $l = 0.001$ | 0.79 | 0.94 | 0.00 | 397 | 0.51 | 0.10 | 4.52 | 2.26 | 0.001 | −732.66 | 30104.74 |
| $c = 0$; $l = 0.001$; $a = 0.1$ | 0.77 | 0.94 | 0.00 | 403 | 0.52 | 0.10 | 5.37 | 2.13 | 0.001 | −742.42 | 30381.13 |

**Replay Task - no-bound**

| Model | $\sigma$ | $\alpha$ | $c$ | $\mu_U$ | $\sigma_U^2$ | $a$ | $b$ | $\lambda$ | $l$ | LLH | $\sum$BIC |
|---|---|---|---|---|---|---|---|---|---|---|---|
| Full | 0.47 | 0.90 | 0.05 | ~ | ~ | ~ | ~ | ~ | 0.012 | −81.13 | 3701.44 |
| $\alpha = 1$ | 0.56 | 1.00 | 0.10 | ~ | ~ | ~ | ~ | ~ | 0.012 | −92.21 | 4030.55 |
| $c = 0$ | 0.48 | 0.90 | 0.00 | ~ | ~ | ~ | ~ | ~ | 0.009 | −82.73 | 3651.38 |
| $l = 0.001$ | 0.50 | 0.91 | 0.06 | ~ | ~ | ~ | ~ | ~ | 0.001 | −82.05 | 3624.39 |
| $c = 0$; $l = 0.001$ | 0.51 | 0.90 | 0.00 | ~ | ~ | ~ | ~ | ~ | 0.001 | −83.64 | 3573.67 |

**Replay task - bound**

| Model | $\sigma$ | $\alpha$ | $c$ | $\mu_U$ | $\sigma_U^2$ | $a$ | $b$ | $\lambda$ | $l$ | LLH | $\sum$BIC |
|---|---|---|---|---|---|---|---|---|---|---|---|
| Full | 0.44 | 0.87 | 0.10 | ~ | ~ | 0.17 | 8.68 | 11.71 | 0.012 | −79.81 | 3991.09 |
| $c = 0$; $l = 0.001$ | 0.48 | 0.88 | 0.00 | ~ | ~ | 0.13 | 8.58 | 15.55 | 0.001 | −82.22 | 3859.24 |
| $c = 0$; $l = 0.001$; $a = 0.1$ | 0.48 | 0.88 | 0.00 | | | 0.10 | 8.91 | 15.88 | 0.001 | −82.38 | 3751.55 |

To compare the Bound and No-bound models in Stage 2. we used five-fold cross validation. The No-bound model had two free parameters: $\alpha$ (temporal bias) and $\sigma$ (inference noise), which were fit to the Same and Less conditions of the Replay task, but tested across all conditions. The Bound model had three free parameters to describe the bound, with the inference noise and temporal bias parameters fixed to those fit to the Same and Less conditions only. In this way, the no-bound model must account for the lack of increased performance in the More condition with the suboptimalities present in the Same and Less conditions, whilst the bound model can limit performance in the More condition in particular by stopping further evidence accumulation. The results of this analysis are presented in the manuscript: the bound significantly improved the fit, mean relative increase in model log-likelihood = 0.048, bootstrapped = 0.001, *Figure 2c* in the main text.

Of additional interest is the pattern of parameters fit to each condition separately, when the model attempts to explain behaviour without a bound. There was little difference in parameters fit to the Same and Less conditions (mean $\sigma_S = 0.48$, $\sigma_L = 0.44$, $Z(19)$ = -1.46, p = 0.15; $\alpha_S = 0.86$, $\alpha_L = 0.78$, $Z(19)$ = 1.38, p = 0.17). The inference noise fit to the More condition significantly increased from the Less condition ($\sigma_M = 0.55$, $Z(19)$ = -2.61, $p_{bonf*4} = 0.036$), but there was significantly reduced temporal integration bias ($\alpha_M = 0.93$, $Z(19)$ = -2.50, $p_{bonf*4}$ = 0. 0496) suggesting observers' performance was worse than predicted by the Same and Less conditions, and they were putting less weight on the more recent evidence. These differences in parameters are consistent with the model trying to mimic bounded evidence accumulation without a bound, providing additional support for the comparison described above.

Stage 3. of the model procedure was to account for the confidence ratings. We compared three processing architectures that span the space from single-channel to dual-channel (*Maniscalco and*

*Lau, 2016*). We took as the null hypothesis a serial processing (single-channel) architecture in which the confidence ratings (Type-II decisions) can be described by the exact same evidence as used to make the perceptual (Type-I) decision. A weaker version of this null hypothesis is that the same suboptimal inference process is used for both perception and confidence, but that the observer can commit to their perceptual decision whilst continuing to monitor additional evidence for evaluating their confidence (a schematic of these processes is shown in *Appendix 1—figure 1*). The average parameter values are shown in *Appendix 1—table 2*, labelled 'Serial' and 'Serial continued' respectively. Note the substantial increase in inference noise ($\sigma$) and reduction in temporal bias ($\alpha$ is closer to 1) when attempting to describe both the perceptual decision and the confidence rating compared to only the perceptual decision (*Appendix 1—table 1*, Replay task – bound, model *c = 0; l = 0.001*). This is indicative of the difficulty of describing both perception and confidence with the same suboptimalities.

At the other extreme is the parallel processing (dual-channel) architecture, in which perception and confidence are computed by independent resources, based on the same sensory input (*Appendix 1—figure 1b*, labelled 'Parallel' in *Appendix 1—table 2*). This is the most computationally expensive description, and provided a lack of parsimony that was only surpassed by a model that attempted to describe confidence ratings with only the inference noise evident from the perceptual decisions.

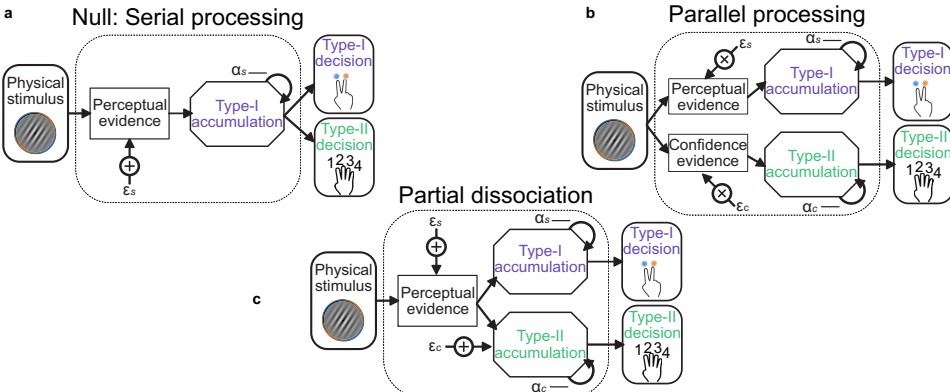

**Appendix 1—figure 1.** Schematic of possible relationships between perceptual (Type-I) and confidence (Type-II) evidence accumulation. (**a**) Same evidence accumulation processes: Type-I (perceptual) and Type-II (confidence) decisions are different responses to the same evidence: each sample of perceptual evidence is disrupted by a sample of sensory noise ($\varepsilon_s$) drawn from a zero-mean Gaussian with standard deviation $\sigma$, and accumulated with a temporal bias described by $\alpha_s$. (**b**) Parallel processing: Type-I and Type-II decisions rely on entirely separate processing of the same physical stimulus: the confidence decision also incurs noise and temporal integration bias (with subscript c), but these may vary independently of the perceptual processing suboptimalities (subscript s). (**c**) Partial dissociation: Type-I and Type-II decisions rely on partially dissociable accumulation of the same evidence.

**Appendix 1—table 2.** Average parameter values for perceptual and confidence behaviour.
Bound parameters with subscript c describe the criteria for confidence ratings, which take the same form as the perceptual decision bound. They have the same minimum and scale, but different rates of decline, such that $\lambda_{c1}$ determines the upper bound on a confidence rating of 1, and the lower bound on a rating of 2. Apart from the 'Serial' and 'Serial continued' models, parameters for perceptual decisions were fixed to those fit in the winning perceptual decision model and the listed parameters affect only the confidence evidence accumulation.

| Model | $\sigma$ | $\alpha$ | $a$ | $b$ | $\lambda$ | $a_c$ | $b_c$ | $\lambda_{c1}$ | $\lambda_{c2}$ | $\lambda_{c3}$ | LLH | $\sum$BIC |
|---|---|---|---|---|---|---|---|---|---|---|---|---|
| Serial | 0.73 | 0.92 | 0.10 | 12.74 | 17.07 | 0.07 | 0.64 | 1.28 | 6.81 | 31.38 | −428.36 | 18275.28 |
| Serial continued | 0.67 | 0.91 | 0.13 | 9.60 | 17.98 | 0.06 | 0.53 | 0.66 | 7.08 | 30.41 | −424.88 | 18135.83 |
| Parallel | 0.76 | 0.90 | ~ | ~ | ~ | 0.01 | 0.58 | 0.18 | 7.51 | 30.68 | −437.25 | 18288.50 |

*Continued on next page*

Appendix 1—table 2 continued

| Model | $\sigma$ | $\alpha$ | $a$ | $b$ | $\lambda$ | $a_c$ | $b_c$ | $\lambda_{c1}$ | $\lambda_{c2}$ | $\lambda_{c3}$ | LLH | $\sum$BIC |
|---|---|---|---|---|---|---|---|---|---|---|---|---|
| Partial - same sigma | 0.00 | 0.89 | ~ | ~ | ~ | 0.06 | 0.47 | 1.03 | 6.77 | 25.92 | −446.41 | 18540.68 |
| Partial - accumulation noise | 0.45 | 0.91 | ~ | ~ | ~ | 0.03 | 0.58 | 0.50 | 7.71 | 31.03 | −421.59 | 17662.25 |
| Partial - read-out noise | 0.12 | 0.90 | ~ | ~ | ~ | 0.02 | 0.52 | 1.85 | 8.63 | 37.39 | −417.94 | 17516.29 |
| Partial - same alpha | 0.12 | 0.88 | ~ | ~ | ~ | 0.02 | 0.52 | 0.98 | 8.22 | 35.16 | −423.02 | 17605.29 |

The intermediate models in this architectural space are the partial dissociation models (*Appendix 1—figure 1c*), which suggest that confidence inherits the same noisy perceptual evidence as the perceptual decision, but may incur some independent suboptimalities. We compared four versions of these models: same $\sigma$ (no additional inference noise); accumulation noise (additional inference noise with each sample of evidence); read-out noise (one additional sample of noise before the confidence response); and same $\alpha$ (the temporal bias affecting the confidence accumulation is the same as that affecting the perceptual accumulation).

In all cases the models were fit to minimise the negative log-likelihood of both perceptual and confidence decisions. The model comparison overwhelmingly favoured the partial dissociation models, and of these, the best description was offered by a model with an independent temporal bias on the confidence evidence accumulation, and additional noise at the read-out stage. We caution against interpreting this result as meaning that there is no additional accumulation noise in the processing of confidence evidence, whilst the models are very similar, it is possible that the read-out noise in this case can additionally capture some noise in setting and maintaining bounds for assigning a rating to the confidence evidence.

The model comparison of Stage 3. just described mainly assumed continued, unbounded accumulation of confidence evidence (with the exception of the strictly serial processing architecture). Stage 4. was to formally compare bounded and unbounded accumulation for confidence evaluations in the same manner as with the perceptual decisions. This time, two versions of the bound were compared: the same bound as perceptual evidence accumulation (the participant could close their eyes after committing to their perceptual decisions and their responses would not change); or an independent bound (the participant can continue to accumulate evidence for confidence decisions after the committing to the perceptual decision, but will eventually stop). As reported in the manuscript, neither bound improved the fit, if anything, adding the bound decreased the log-likelihood of the model (same bound: relative improvement with bound = −0.007, bootstrapped p = 0.11, uncorrected; independent bound: relative improvement = −0.014, p = 0.022, Bonferroni corrected for two comparisons; *Figure 2c*, in the main text). This reflects the fact that even a very high bound affects the shape of the accumulation trace, which will harm the fit when behaviour is not affected by a bound.

In summary, this computational modelling procedure suggests a partial dissociation in the processing for perception and confidence. In the Replay task, perceptual decisions were best described by bounded evidence accumulation, enabling observers to commit to decisions before the sequence of presented samples finishes. The confidence ratings required additional noise and reduced temporal integration bias compared to the suboptimalities affected the perceptual decisions. These differences were best described by the partial dissociation architecture where confidence received the same noise samples of evidence as the perceptual decision, though they are accumulated differently. In addition, model comparison suggested confidence evidence accumulation continued to the end of the sequence, even in cases of premature commitment to the perceptual decision. The results of these comparisons replicate the results of *Balsdon et al., 2020*, with the exception of the confidence noise comparison: here we find evidence in favour of read-out noise, whereas the previous analysis found the models indistinguishable.

## Appendix 2

## Model Simulation

The computational model comparison suggested a partial dissociation in the evidence used to make perceptual decisions and confidence evaluations. We compared the evidence underlying the observers' perceptual decisions and confidence ratings by simulating the winning computational model. For each trial, 10,000 samples of noise per decision update were randomly sampled from the Gaussian distribution describing the observer's inference noise. These were combined to give 10,000 simulated evidence traces per trial. The first 1,000 simulated evidence traces that agreed with the observer's response on that trial were taken to measure the median evidence trace (or, the process was repeated until 1,000 adequate simulated evidence traces were drawn, up to 100 repeats). *Appendix 2—figure 1a* demonstrates this process for one example trial of one observer. For the perceptual evidence (*Appendix 2—figure 1a*, left) simulated evidence traces that agreed with the observer's response are those that reach the respective decision bound before the opposing decision bound, or reach no bound but show evidence in favour of the response by the final sample. It was assumed that once the evidence reaches the bound, that evidence is maintained until the response. For the confidence evaluation (in the example, a confidence rating of 3), the final evidence had to be between the confidence rating bounds to agree with the observer's confidence decision (after the final sample of additional noise – which is why a few samples in *Appendix 2—figure 1a*, right, exceed the bounds). The median evidence was compared to the ideal evidence (green lines of *Appendix 2—figure 1a*).

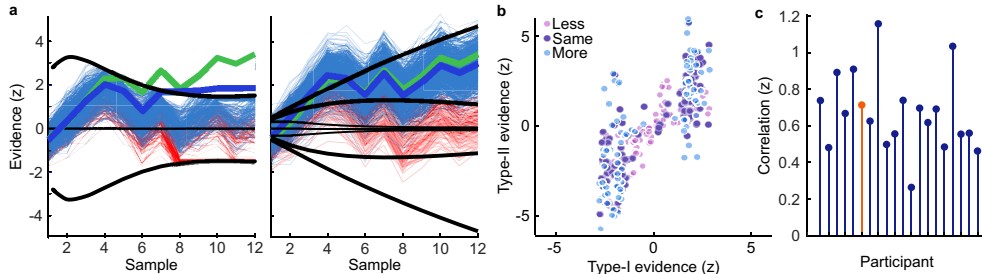

**Appendix 2—figure 1.** Model simulation of accumulated evidence for perceptual and confidence decisions. (**a**) Example trial from one observer showing simulated evidence traces agreeing with the observer's response (blue) and a sample of example traces which did not agree (red). The perceptual decision is shown on the left. An evidence trace was taken to agree with the observer's decision if the corresponding bound was reached prior to the opposing bound, or if no bound was reached but the final accumulated evidence was in favour of the chosen option. The median evidence trace (thick blue line) was calculated assuming the evidence that reached the bound early was maintained until the response was entered. For the confidence rating (right) we compared the median evidence from traces where the final accumulator (plus one additional sample of noise) agreed with the observer's confidence rating. We examined the difference from the ideal accumulated evidence (thick green line) relative to the likelihood of the observers' rating given all simulated evidence traces. (**b**) Median final simulated accumulated evidence for the perceptual decision (abscissa), and the confidence decision (ordinate) for all trials of the example observer, colours indicate the condition. (**c**) Correlation (Fisher transformed z) between perceptual and confidence evidence for each observer. The example observer is highlighted in orange.

The estimated inference error (used in Appendix 7) scaled the difference between the median consistent evidence and the ideal evidence by the probability of the response given all samples, to estimate the relative deviation of the observers' internal evidence from the optimal observer's evidence. This estimate of the error is quite imprecise: the median trace tends to be quite close to the ideal, even though any one of the traces (which reflect much larger error) could have described the internal evidence of the observer. *Appendix 2—figure 1b* shows the predicted final accumulated evidence for the perceptual (Type-I) compared to the confidence (Type-II) decision for the same example observer. The evidence is strongly correlated but there are substantial deviations, because of the additional noise, different temporal bias, and continued accumulation for the confidence

decision, especially in the More condition (light blue). The example observer is a more extreme case because of the relatively strong bound on perceptual evidence accumulation. The (Fisher transformed) correlation for each observer is shown in *Appendix 2—figure 1c*. For many observers there are substantial differences between the median simulated evidence consistent with the perceptual and confidence responses, meaning the simulated evidence could be useful in distinguishing representations important for perception vs. confidence.

# Appendix 3

## Confidence behaviour

Proportion correct increased with increasing confidence, reflecting the observers' ability to use their confidence ratings to discriminate correct from incorrect responses (*Appendix 3—figure 1a*). Observers appeared to be monitoring the decision evidence to make their confidence ratings, as opposed to some proxy for confidence such as the number of samples they were shown (*Appendix 3—figure 1b and c*).

We required a single-trial measure of confidence precision for identifying the key neural processes underlying the computation of confidence. To do so, we compared observers' responses to an optimal observer. The optimal observer perfectly accumulates all presented evidence and assigns ratings to equally partition the evidence for their perceptual decision. To simplify, we split trials by the median evidence for the chosen category, where the optimal observer gives a high confidence rating (3 or 4) to those trials with greater than the median evidence, and a low confidence rating (1 or 2) to those with less than the median evidence. We labelled trials as 'suboptimal confidence' when the observer's confidence response disagreed with the response of this optimal observer. This trial labelling is demonstrated for two example observers in *Appendix 3—figure 1d*. We reasoned that on suboptimal confidence trials the internal evidence of the human observer was less likely to be close to the optimal presented evidence, and the neural representation of the optimal presented evidence should be less precise in neural circuits that actually represent this suboptimal confidence evidence. That this measure of confidence precision does capture the suboptimalities in confidence evaluation is confirmed by the significant increase in model estimated confidence error on suboptimal confidence trials (Wilcoxon sign rank test: $Z(19) = 3.85$, $p < 0.001$; *Appendix 3—figure 1e*).

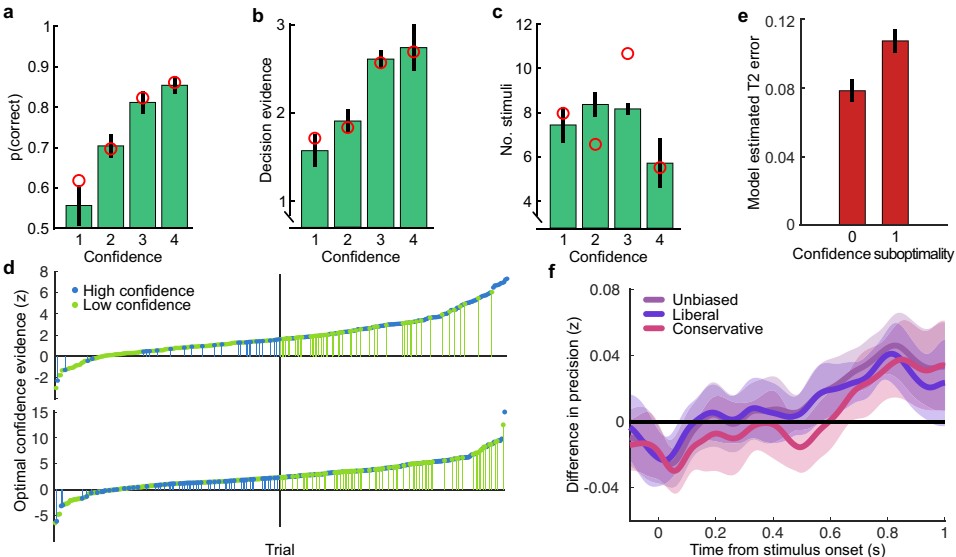

**Appendix 3—figure 1.** Confidence behaviour. (**a**) proportion correct (in the perceptual decision) by confidence rating. (**b**) Decision evidence (based on the presented samples) by confidence rating. (**c**) Number of samples presented by confidence rating. In all plots, error bars show 95% within-subject confidence intervals. Red circles show the predictions of the best fitting confidence model (Appendix *1*). (**d**) Confidence responses of two observers (top and bottom panels) on all trials sorted by the confidence evidence of the optimal observer. The median confidence evidence (shown by a black vertical line) defines an optimal confidence observer whose confidence above this median are rated high. Observers' high confidence ratings are shown in blue and low confidence ratings in green. Suboptimal confidence ratings, where human and optimal confidence observers do not match, are indicated with small vertical segments (green for Type-II misses and blue for Type-II false alarms). Negative confidence evidence corresponds to incorrect perceptual decisions. The observer shown on top clearly has fewer suboptimal responses compared with the observer below, and the

*Appendix 3—figure 1 continued on next page*

*Appendix 3—figure 1 continued*

frequency of suboptimal responses decreases further from the median. (**e**) Model estimated confidence error by confidence rating suboptimality (0 = the observer's confidence rating was the same as the optimal observer, 1 = suboptimal confidence rating). (**f**) The effect of response bias on the analysis of suboptimal confidence in the EEG representation of accumulated evidence. Observers' confidence ratings were compared to an unbiased optimal observer (purple), and two biased (but otherwise optimal) observers, who respond with high confidence on 35% and 65% of trials (making the human observers relatively more liberal and conservative with their response strategy in comparison). Thick lines show the within-subject difference in precision (Fisher transformed correlation) between trials where the human observers' confidence ratings correspond to the (un/biased) optimal observer and suboptimal confidence ratings. Shaded regions show the 95% between-subject confidence intervals on the difference.

In this way, observers' confidence is assessed relative to a "super-ideal" observer, who has perfect access to the presented evidence (*Mamassian and de Gardelle, 2021*). Theoretically, observers' confidence should be assessed relative to the internal evidence for their perceptual decision, that is, relative to the evidence based on suboptimal inference (afflicted by noise and temporal integration biases). However, the single-trial estimates of the internal evidence for perceptual decisions, based on model simulations, were relatively imprecise (see Appendix 2), and could also introduce systematic errors from the model assumptions, making this estimate of the internal evidence unappealing for the purpose of assessing confidence. Moreover, the goal of this measure was to compare observers' confidence ratings to the neural representation of the accumulated evidence, which was also assessed relative to the optimal evidence. We therefore chose to assess confidence ratings relative to the optimal observer in the same way that neural responses were assessed relative to optimal, though this ignores the fact that some suboptimality is actually inherited from perceptual decision processes.

A second important consideration with this measure is that it is affected by confidence bias. There are three types of biases that could affect confidence ratings: first, a response bias to enter a certain response irrespective of the evidence; second, a miscalibration bias such that ratings mean different things to different observers (the same value of evidence will be given a rating of 4 for one observer and 3 for another, for example); third, a miscalling bias such that perceptual evidence is relatively exaggerated or diminished in the assessment of confidence. All these biases mean that the same internal perceptual evidence could result in systematically different confidence ratings across observers, and observers could report on average higher or lower confidence despite similar perceptual performance and precision in representing the internal evidence for evaluating their confidence.

Taking an average proportion of suboptimal confidence ratings and comparing across observers would result in observers of similar ability having different scores simply because of biases in how they implement the confidence rating responses: greater biases will increase average proportion suboptimal. Importantly, this single-trial measure of confidence was not used for this purpose. Rather, it was compared to neural activity during the process of accumulating evidence for the perceptual decision and confidence evaluation. We expect that biases that are not of interest for the computation of confidence (in particular, response bias and miscalibration bias) are incorporated at a later stage, when the confidence evaluation is converted into a rating for executing the response. The biases will only reduce the sensitivity with which a trial labelled as suboptimal truly reflects internal evidence that differs from optimal, reducing our ability to identify neural processes underlying confidence computation. This is simulated in *Appendix 3—figure 1f*, where a relative bias is introduced by assessing human confidence ratings to a biased optimal observer (who responds on 65% of trials with high confidence – making the human observers relatively more liberal, or 35% high confidence – making the human observers more conservative). The general trend for the difference between confidence ratings that match the (biased) optimal observer and those that are suboptimal remains the same, though the bias reduces the difference.

## Appendix 4

### Classical EEG analyses

To link back with the previous literature, we present here two more classical EEG analysis approaches, examining the modulations of EEG amplitude around the time of the response. In *Appendix 4—figure 1a*, we show the Lateralised Readiness Potential (LRP; difference in microvolts between the average of electrodes [C1, C3], and [C2, C4], signed by response hand; *Deecke et al., 1976*). The data are unfiltered with the exception of the pre-processing, and baselined using the 100 ms before the onset of the first stimulus of each trial. There was a significant difference in the LRP between the Less and More conditions of the Replay task from just after the response (the first cluster from 32 to 196 ms; $t_{ave}(19) = -3.57$, $p_{cluster} < 0.002$, *Appendix 4—figure 1a*, top). There were also differences based on perceptual decision accuracy (from −84 ms to 652 ms around the response, with the largest difference just after the response, $t_{ave}(19) = 2.81$, $p_{cluster} < 0.002$; *Appendix 4—figure 1a*, middle). There was no significant difference in the LRP between trials with high confidence (ratings of 3 and 4) and low confidence (ratings of 1 and 2; *Appendix 4—figure 1a*, bottom).

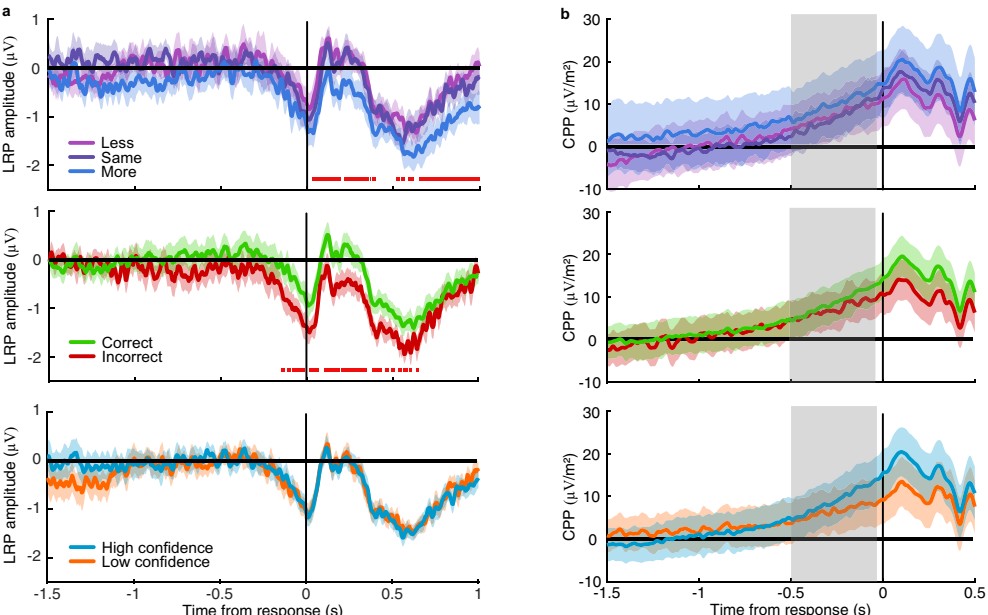

**Appendix 4—figure 1.** Amplitude modulations with task variables. (**a**) The Laterised readiness potential by condition (top), perceptual decision accuracy (middle) and reported confidence (bottom). Horizontal red lines mark significant differences in amplitude. (**b**) Central Parietal Positivity, with the same comparisons. Shaded regions show 95% within subject confidence intervals, and the region of slope comparison for the CPP is highlighted in grey.

We also computed the Central Parietal Positivity (CPP; *O'Connell et al., 2012*) which has previously been shown to reflect perceptual evidence accumulation. We followed the methods presented in *Kelly and O'Connell, 2013*: data were lowpass filtered at 45 Hz with no highpass filter, and converted to current source density (*Kayser and Tenke, 2006*). As with the LRP, a baseline was taken from the 100 ms before the onset of the first stimulus of each trial. The slopes of the CPP (a linear fit from −500 to −50 ms) showed no significant differences across all conditions ($F(1,19) = 2.15$, p = 0.14). We observed a significantly greater slope for correct compared to incorrect decisions ($t(19) = -2.86$, p = 0.01), and an even greater difference between high and low confidence trials ($t(19) = -3.24$, p = 0.004). This is consistent with the literature suggesting the CPP traces the internal evidence for the perceptual decision, however it is difficult to disambiguate how this signal may differentially contribute to perceptual decisions and confidence evaluations.

## Appendix 5

### Response classification

A linear discriminant analysis was used to classify the perceptual decision response based on the spectral power of band-limited EEG signals in epochs locked to the time of the response. The spectral power across frequency tapers from 1 to 64 Hz with 25% spectral smoothing was resolved using wavelet convolution implemented in FieldTrip (*Oostenveld et al., 2011*). The epochs were then clipped at −3 to 1 s around the time of entering the perceptual decision response. We first trained and tested at each frequency taper at each time point in the Free task (*Appendix 5—figure 1a*). Classifier performance was measured as the area under the curve (AUC). The power in frequency bands between 8 and 32 Hz yielded the most accurate classification performance. The difference in the average power across these frequency bands between −0.5 and 0.5 s around the time of the response for right- and left-handed responses showed a clear lateralisation over central and parietal electrodes (*Appendix 5—figure 1b*). Training and testing at each time point in each condition of the Replay task showed a similar pattern to the Free task, with reliable classifier performance from around −0.5 to 0.5 s around the response (*Appendix 5—figure 1c*). Training and testing within each condition of the Replay task resulted in a larger between-subject error, likely because there are only 100 trials per condition. In the main text, we present a cross-classification analysis where the classifier is trained on the Free task, and tested on each condition in the Replay task, which more directly examines when the signals relevant for entering a response (based on the Free task) emerge during the lead up to the response in each condition of the Replay task.

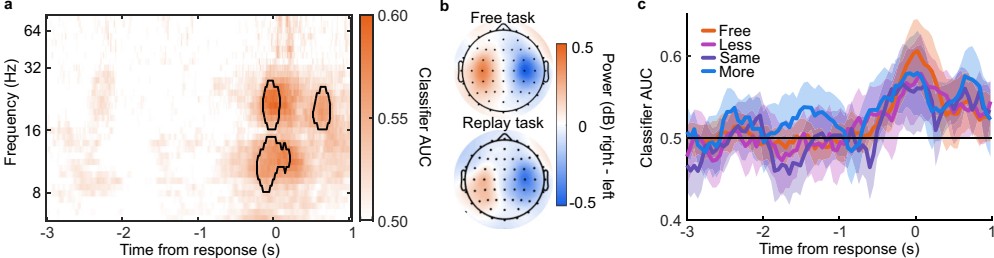

**Appendix 5—figure 1.** Response Classification analysis. (**a**) Classifier AUC training and testing at each time point (abscissa) based on the power (dB) in each frequency band (ordinate). Clusters where average performance is greater than 3.1 standard deviations (99% confidence) from baseline (0.5) are circled in black. (**b**) Scalp map of the difference in power for right- compared to left-handed responses averaged over 8 to 32 Hz and −0.5 to 05 s around the response. (**c**) Classifier performance (AUC) training and testing at each time point, in each condition of the Replay task and in the Free task.

## Appendix 6

### Encoding variable regression

Linear regression was used to examine the representation of encoding variables in the EEG signals. First, regression weights ($\widehat{W}$) were computed using ridge regression of the encoding variables ($C$, an $n \times 1$ matrix) on the EEG signals ($D$, an $n \times m$ matrix, where $m$ is the number of EEG signals, and $n$, the number of epochs)

$$\widehat{W} = \left(D^T D + \lambda I\right)^{-1} D^T C \qquad (A4)$$

The regularisation parameter, $\lambda$, was set to 1, where $I$ is the identity matrix. Weights were computed on 90% of the epochs, and used to predict the encoding variables on the other 10% (10-fold cross validation) simply as: $\widehat{C} = D * \widehat{W}$. The precision of the prediction was calculated as the correlation between $\widehat{C}$ and $C$, standardised using a Fisher transformation.

Three different encoding variables, $C_\theta$, $C_\ell$, and $C_z$, were examined (*Appendix 6—figure 1a*): the stimulus orientation ($C_\theta = \pi - |\theta_n|$), the momentary decision update ($C_\ell = |\ell_n| = |\kappa cos(2(\theta_n - \mu_1)) - \kappa cos(2(\theta_n - \mu_2))|$), and the accumulated evidence ($C_z = z_n = \sum_{N=1}^{n} \ell_N$, signed by the response). These variables are not entirely independent: There is a weak correlation between the stimulus orientation and the momentary decision update ($r = 0.03$), and a weak correlation between the momentary decision update and the accumulated evidence ($r = 0.09$). In addition, the accumulated evidence is strongly correlated over samples ($r = 0.92$ at n+1, and $r = 0.85$ at n+2). The cross-correlations are shown in *Appendix 6—figure 1c*.

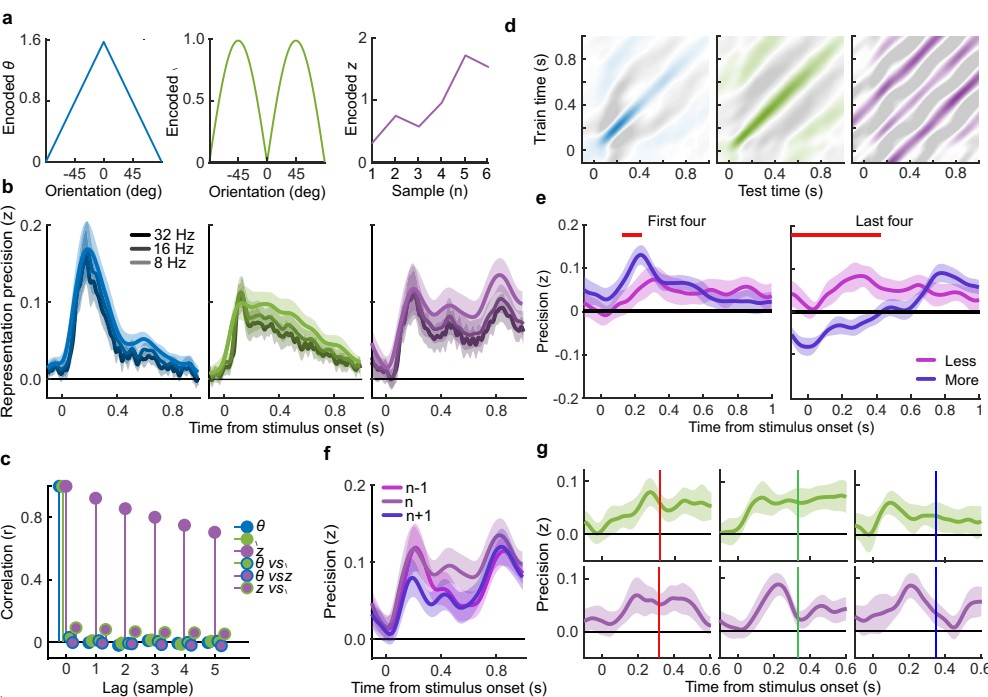

**Appendix 6—figure 1.** Encoding variable regression. (**a**) Encoded variables used to regress EEG signals. The encoded orientation ($C_\theta$, left) and encoded momentary decision update ($C_\ell$, middle) were dependent on the orientation presented to the observer. The encoded accumulated evidence ($C_z$) varied over all presented orientations in a trial, the figure on the right shows only one example. (**b**) Representation precision of encoding variables using different low-pass filters. (**c**) Cross correlation between encoding variables over consecutive samples. (**d**) Temporal generalisation of representations: the regression weights were calculated on EEG signals at each time point and precision was tested across time. Colour scales are relative to the maximal precision, with zero

*Appendix 6—figure 1 continued on next page*

*Appendix 6—figure 1 continued*

precision in white and negative in grey (a sign flip of the regression weights). (**e**) Representation precision of the accumulated evidence for the first (left) and last (right) four stimuli of the Less and More conditions. Shaded error bars show the 95% within subject confidence intervals, red horizontal bars mark cluster corrected significant differences between conditions. (**f**) Representation precision of the previous (n-1), current (n) and future (n+1) accumulated evidence, based on the EEG signals locked to the current epoch. (**g**) Representation precision of the momentary decision update (top) and the accumulated evidence (bottom) for epochs separated by the timing of the subsequent stimulus, shown in coloured bars (317 ms, red, left; 333 ms, green, middle; and 350 ms blue, right).

The EEG signals in D were low-pass filtered and decomposed into real and imaginary parts using a Hilbert transform. Regression precision was first calculated using the signals from all electrodes ($m$ = 128) separately for each time-point in the stimulus-locked epochs. Initial analysis showed a low-pass cut-off of 8 Hz was appropriate to decrease noise whilst maintaining precision (*Appendix 6—figure 1b*). The previous literature has shown similar results (*Salvador et al., 2020*).

Temporal generalisation of the representation of encoding variables was tested by computing weights at each time point and testing the predicted encoding variables across time (*Appendix 6—figure 1d*). Though the representation of the momentary decision update is maintained for a relatively longer duration than the representation of stimulus orientation, there is little temporal generalisation, suggesting the representation in the EEG signals evolves over time. This is also the case for the representation of accumulated evidence, however, there are also strong off-diagonals in the temporal generalisation matrix. This is likely because of the strong correlation across consecutive samples (*Appendix 6—figure 1c*).

The precision of the representation of accumulated evidence was compared across the Less and More conditions for the first four and the last four stimuli (*Appendix 6—figure 1e*). As reported in the main text, representation precision was substantially attenuated for the last four stimuli of the More condition. This was not the case for the first four samples, where decoding precision in the More condition was briefly (from 132 to 244 ms) greater than in the Less condition ($t_{ave}(19)$ = 3.67, $p_{cluster}$ < 0.001).

Given the sustained precision of decoding accumulated evidence over time, and the strong correlation between consecutive samples, it is curious that the measured precision does drop to baseline at the start of the epoch. That the same pattern is found when decoding sample n-1 and sample n +1 based on the epoch at sample n (*Appendix 6—figure 1f*) suggests that the onset of the stimulus is disrupting the ongoing representation (or at least, our ability to measure it). Furthermore, this decrease in performance is not seen in the temporal generalisation matrix, where the off-diagonal is not aligned with the onset of successive samples (due to the jitter in stimulus presentation timing). Comparing precision between groups of epochs where the timing of the subsequent sample is aligned (*Appendix 6—figure 1g*; red 317 ms, green 333 ms, blue 350 ms) suggests there could be an interaction between the timing of ongoing updates and the precision of the representation of the accumulated evidence (but not the momentary decision update). This could be of interest for future research.

# Appendix 7

## Cluster modelling

Cluster modelling was used to isolate contiguous signals in space (electrode location) and time, where the precision of the representation of optimal accumulated evidence systematically varied how closely the internal representation of evidence matched optimal, based on whether behavioural responses matched the optimal observer. We assume that responses that do not match the optimal observer are based on evidence that deviates further from the optimal evidence. Neural signals that reflect the internal evidence of the observer should also deviate further from the optimal evidence used in the regression on these trials, and be closer to the optimal evidence on trials where the observers response matched the optimal observer. Clusters were isolated using a multivariate Bayesian scan statistic (*Neill, 2011*; *Neill, 2019*). This statistic was calculated based on the log-likelihood ratio of the alternative hypothesis (that representation precision depends on the internal evidence of the observer) against the null hypothesis (that any difference in representation precision is due to measurement noise alone, which is independent across epochs). It is assumed that the neural signals reflect the input (cumulative presented evidence) with added measurement noise ($N_m$) and, when the neural signals are relevant for behaviour, inference noise ($N_i$) that reflects the deviations from the optimal evidence in the internal representation of the observer

$$Y_{out} = Y_{in} + N_i + N_m \tag{A5}$$

Where the two sources of noise are assumed to be gaussian distributed ($N(0, \sigma^2)$). The total measured correlation ($r_T$) between $Y_{in}$ and $Y_{out}$ is a function of the additional noise (where $Y_{in}$ is normalised)

$$r_T = \frac{1}{\sqrt{2 + \sigma_i^2 + \sigma_m^2}} \tag{A6}$$

When the observer's decision does not match the optimal decision their internal representation of the accumulated evidence is likely to be further from the optimal value, resulting in a weaker correlation between the internal representation and the presented evidence. Therefore, when we split based on behaviour, we expect that on average there is greater inference noise on incorrect trials than correct trials. The correlation over all samples can be described as

$$r_T = \frac{1}{\sqrt{2 + p(I)\sigma_{iI}^2 + p(C)\sigma_{iC}^2 + \sigma_m^2}} \tag{A7}$$

where $p(I)$ is the observed probability of a decision that does not match the optimal observer, and $p(C)$, a decision that corresponds to that of the optimal observer. The null hypothesis is that the neural signal is not relevant for behaviour, specifically, signals on suboptimal trials do not reflect additional inference noise. Any difference in the correlation is due to variance in the measurement noise,

$$H_0: \sigma_{iI} = \sigma_{iC} = 0 \tag{A8}$$

The alternative hypothesis is that the neural signals are relevant for behaviour, reflecting the greater variance from optimal on trials where the observer makes a decision that does not match the optimal decision,

$$H_1: \sigma_{iI} > \sigma_{iC}, \text{ or } \sigma_{iI}^2 = (\sigma_{iC}^2 - x) \text{ where } x > 0 \tag{A9}$$

The difference in the inference noise is limited by the total variance

$$p(I)(\sigma_{iI}^2) + p(C)(\sigma_{iI}^2 + x) = \frac{1}{r_T^2} - 2 - \sigma_m^2 \tag{A10}$$

Solving for $\sigma_{iI}^2$ (since $p(C) + p(I) = 1$):

$$\sigma_{iI}^2 = \frac{1}{r_T^2} - 2 - \sigma_m^2 - p(C)x \tag{A11}$$

If we consider the correlation between the neural representation and the presented evidence on trials with optimal and non-optimal responses separately (for simplicity, let $R = \frac{1}{r_T^2}$),

$$r_I = \frac{1}{\sqrt{R - p(C)x}} \tag{A12}$$

$$r_C = \frac{1}{\sqrt{R - p(C)x - x}} \tag{A13}$$

Setting a uniform prior on the ratio of inference and measurement noise, results in a linearly descending prior on $x$

$$p(x) = \frac{R - 2 - p(I)x}{\int_0^{(R-2)/p(I)} R - 2 - p(I)x \, dx} \tag{A14}$$

We actually measure the difference in the Fischer transform of the correlation

$$z_C - z_I = 0.5 \log\left(\frac{(1 + r_c)(1 - r_I)}{(1 - r_c)(1 + r_I)}\right) \tag{A15}$$

Since $r_c$ and $r_I$ are independent of the assumed measurement noise, there is one $x$ that corresponds to a measured difference $z_C - z_I$, given the overall correlation $r_T$.

For each participant, for each electrode, at each time-point, the prior on $\sigma_m^2$ for $H_0$ is calculated by permuting the data labels (accurate vs inaccurate behavioural responses). The probability of the data given $H_0$ and $H_1$ are calculated as above and used to compute the loglikelihood ratio

$$LLR = log\left(\frac{p(D|H_1)}{p(D|H_0)}\right) \tag{A16}$$

The clusters are identified using the Fast Subset Sums procedure: The loglikelihood ratios are summed across participants, for each electrode and time-point. We then find small clusters by thresholding the log posterior odds ratio

$$POR = LLR + log\left(\frac{p(H_1)}{p(H_0)}\right) \tag{A17}$$

where the prior $p(H_1)$ is set to 0.05. The cluster with the largest LLR (summed across electrodes and time points) is then expanded by continuing to add the largest neighbour and the new log prior ($p(H_1) = 0.05/n$), where $n$ is the size of the cluster, whilst the POR remains in favour of $H_1$. This is repeated until all clusters with evidence in favour of $H_1$ have been identified.

## Appendix 8

### Estimating single-sample confidence inference error

We aimed to examine the neural processes that are important for the representation of the decision evidence for computing confidence. To do so, we explored the source(s) of the activity reflecting the neural representation of the accumulated evidence in the clusters of signals identified as relevant for confidence evaluations. We use the representation from the cluster as an estimate of the internal evidence the observer uses to make their confidence evaluations. The cluster inference error is the absolute difference between the predicted value (on each sample) and the optimal value given the presented evidence. We take this as an estimate of the inference error of the observer at the sample level. This estimate is likely substantially affected by measurement noise, however, we do not expect measurement noise to be systematically driven by a specific source, especially not across subjects. Noise Min and Noise Max epochs were selected by taking the top and bottom quartiles of epochs sorted by this estimate of inference error.

A separate estimate of the inference error was obtained by simulating the computational model (*Appendix 8—figure 1a* shows the process of obtaining these estimates and their mutual reliance on the input stimulus variables and the behavioural output). This computational model estimate also has its drawbacks: It is relatively imprecise, given the large range of errors that are consistent with the observers' behavioural responses (see Appendix 2); and is based on the assumptions of the model. By examining these two estimates, we avoid relying on the same set of assumptions throughout the analysis. As reported in the **Results** section, the estimate of the single-sample inference error from the cluster representation was significantly correlated with the single-sample inference error estimated from the computational model of confidence ratings ($t(19) = 5.12$, $p < 0.001$), and this correlation was significantly greater than the error estimated from the model of perceptual decisions alone ($t(19) = 2.62$, $p = 0.017$). This correlation between these estimates suggests that they do tap into the suboptimal inference of the observer.

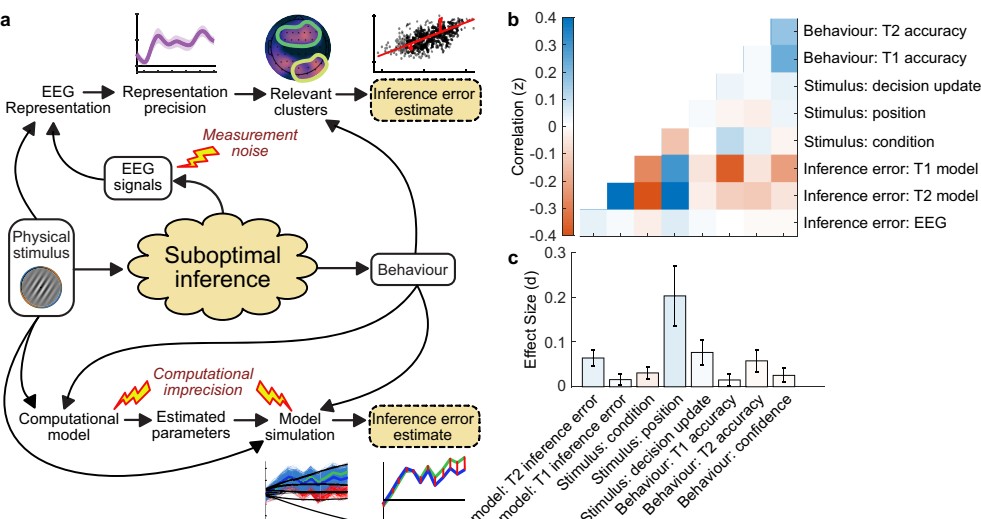

**Appendix 8—figure 1.** Estimating inference error. (**a**) Two approaches to estimate inference error. It is assumed the observer's behaviour is based on a suboptimal inference over the physical stimulus. We do not have access to the single-sample inference error, but can estimate it using the measured variables: the physical stimulus properties, the behaviour, and the EEG signals. Two approaches are outlined: The EEG inference error estimate, which relies on the error of the representation of the accumulated evidence, in clusters where the precision of the representation is related to suboptimal behaviour; and the model error, which relies on simulating the processing of the evidence based on the fitted model parameters, and taking the median of simulated traces which concur with the observer's response. (**b**) Correlation between variables measured from behaviour, the stimulus input, and the estimated inference error. (**c**) Effect size on the difference between Noise Min and Noise Max epochs.

*Appendix 8—figure 1b* shows the correlation of these estimates of the inference error and different variables related to the stimulus presentation and behaviour, averaged across subjects. We also examined the average absolute effect size of the within subject difference between different variables dividing trials by Noise Min and Noise Max epochs is shown in *Appendix 8—figure 1c*. There was a larger effect on confidence inference error ($d = 0.06$) than perceptual inference error ($d = 0.02$), from the model estimate. There were some effects on stimulus variables: a small effect of condition (More vs Less, $d = 0.03$), a large effect on sample position in the sequence (Noise Min epochs tended to correspond to earlier samples, $d = 0.2$), and an effect on decision update (Noise Min epochs tended to correspond to smaller momentary decision updates, $d = 0.08$). The effects on behaviour were largest for confidence accuracy ($d = 0.06$), with limited effect on perceptual accuracy ($d = 0.02$) and confidence rating (Noise Min epochs were somewhat more associated with high confidence ratings, $d = 0.03$).

## Appendix 9

### Regions of interest

Regions of interest were selected based on the previous literature. Specifically, *Herding et al., 2019* found subjective evidence to modulate activity in the superior parietal cortex; *Gherman and Philiastides, 2018* found correlates of confidence encoding in the ventro-medial prefrontal cortex (overlapping with the MindBoggle orbitofrontal cortex coordinates), whilst *Graziano et al., 2015* examined ROIs in the anterior cingulate cortex, orbitofrontal cortex, temporal lobe, posterior parietal cortex, and occipital cortex. We chose to use ROIs defined by MindBoggle (*Klein et al., 2017*) that corresponded to similar regions: lateral occipital cortex, superior parietal cortex, orbitofrontal cortex (combining medial and lateral partitions), rostral middle frontal cortex, and initially the anterior cingulate cortex (combining rostral and caudal partitions; *Appendix 9—figure 1b*). These regions do not necessarily map on to regions of the greatest current density (*Appendix 9—figure 1a* shows the current density over time for the Noise Min epochs). The results of the anterior cingulate cortex were similar to the neighbouring orbitofrontal region, so we decided not to present this in the manuscript for simplicity. We show the results in *Appendix 9—figure 1c*, for left and right hemispheres separately (statistical analyses were performed on the average).

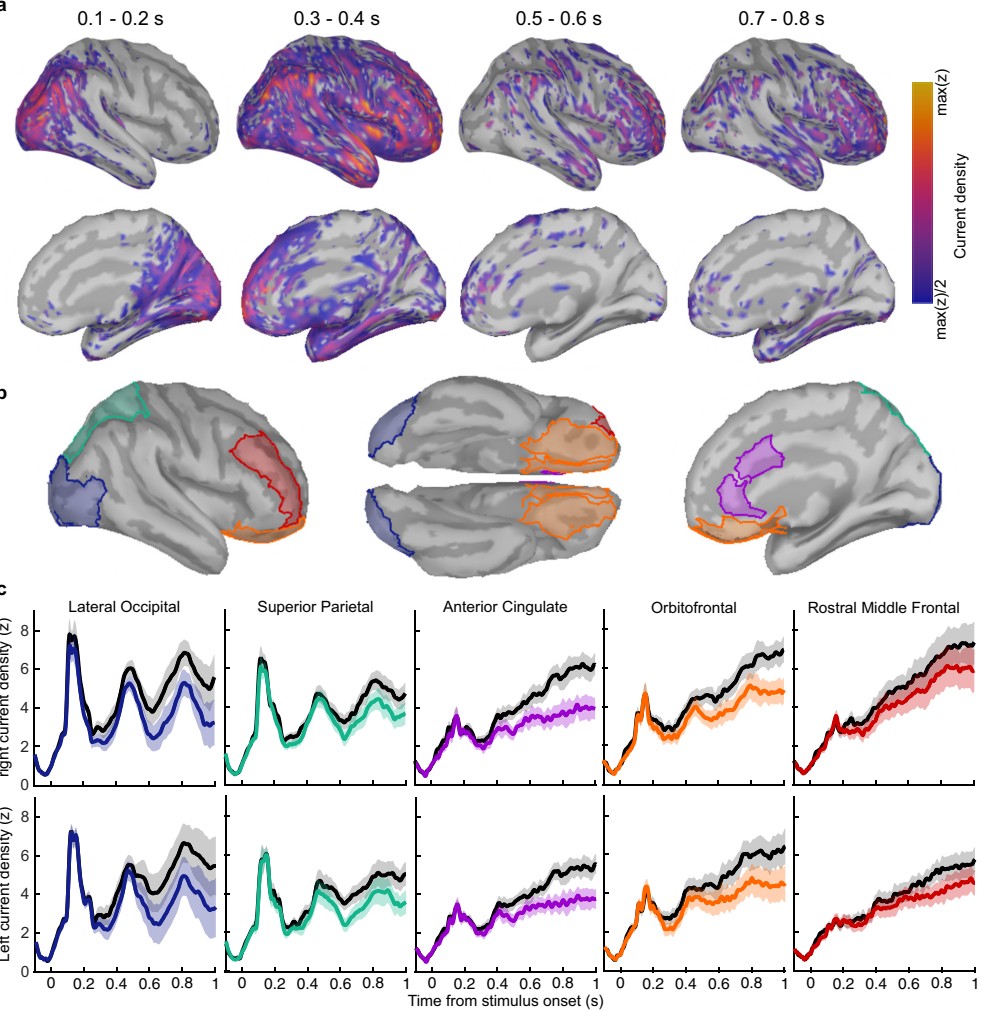

**Appendix 9—figure 1.** Regions of interest and corresponding current density. (**a**) Average rectified normalised current density in Noise Min epochs for the corresponding time windows, filtered above the half-maximum amplitude (**b**) Regions of interest based on Mindboggle coordinates. (**c**) Average normalised rectified current density in the right (top) and left (bottom) hemispheres. Noise Min

*Appendix 9—figure 1 continued on next page*

*Appendix 9—figure 1 continued*

epochs are shown coloured, Noise Max in black, with shaded regions showing the 95% within-subject confidence interval.

