## [Decision Letter]

**Acceptance summary:**

This paper is of interest to neuroscientists and psychologists working on perceptual decision-making and metacognition. Using a novel task varying the timing of covert decisions, together with sophisticated computational modelling, allowed identifying neural correlates of latent states related to confidence. The conclusions are in line with other papers identifying a dissociation between brain activity supporting performance and confidence, but provide a novel lens through which to understand these differences by focusing on confidence noise.

**Decision letter after peer review:**

Thank you for submitting your article "Separable neural signatures of confidence during perceptual decisions" for consideration by *eLife*. Your article has been reviewed by 2 peer reviewers, including Redmond G O'Connell as the Reviewing Editor and Reviewer #1, and the evaluation has been overseen by Michael Frank as the Senior Editor. The reviewers have discussed their reviews with one another, and the Reviewing Editor has drafted this to help you prepare a revised submission.

Essential Revisions (for the authors):

1) The study needs to be more clearly framed with respect to the previous literature. Extensive neuroimaging and lesion work has already provided compelling evidence that perceptual decisions and metacognition rely on distinct neural circuits. In addition, previous modeling work that has provided a joint account of first- and second-order behaviour has assigned a key role to post-decision processing (whether within same or different circuits). The authors should clarify their hypotheses and how they relate to this previous work.

2) Substantial rewriting of the Results is necessary. A clear rationale should be provided for each analysis step and the results of those analyses need to be linked back to the study hypotheses.

3) Consider whether a simpler model-free analysis (e.g. check whether confidence resolution is higher for More vs Less trials) could be done to validate the key claim of confidence/performance dissociations (see Reviewer 2 comment)

4) Check impact of high-pass filtering and current source density on CPP signals (or consider removing those signals from the analysis if they are not speaking to the study hypotheses).

*Reviewer #1 (Recommendations for the authors):*

Some substantial rewriting of the Results section is required to clarify the rationale for each analysis without requiring that the reader read all of the Supplemental Materials in order to follow the narrative. In several instances it is not clear why the analysis is being run or what we should take from it. A prominent example is the ERP analysis where the choice of signals or their relevance to the hypothesis is not explained and the relevance of the reported results to those hypotheses is similarly not clarified. It is not clear why only the More and Less conditions are compared or why post-response time-points are considered relevant. Additionally the purpose of the decoding analysis is left obscure and it is not clear why 8-32Hz is the focus Throughout I think a lot more hand holding is required for the reader to follow why each analysis is being run and how exactly it relates to the stated hypotheses.

I also have a few more specific comments regarding particular aspects of the results.

If the CPP analyses are deemed important to the overall story then the use of a 0.5Hz filter may be problematic as it is likely to attenuate a slow building signal like the CPP. This could conceivably lead to some issues when comparing Less/Same/More trials where the CPP may build over different durations. The authors might also consider applying CSD transformation which has been shown to reduce interference from overlapping fronto-central negativities (e.g. Kelly and O'Connell 2013, J Neurosci).

One of the key observations is that the precision of neural representations of accumulated evidence drop towards the end of the More trials but there is no such dropoff in the decision update representation. I am struggling to understand how this might arise and it would be helpful if the authors could provide an explanation.

The authors report that there is no significant effect of More vs Same on choice accuracy but the figure suggests that there is a substantial numerical difference in the expected direction. Bayes factors should be provided to quantify the evidence in favour of the null hypothesis.

*Reviewer #2 (Recommendations for the authors):*

I found myself wondering in lots of places here whether a simpler model-free analysis could have been done to validate the key claim of confidence/performance dissociations in behaviour and neural activity. For behaviour, would it not be possible to eg check whether confidence resolution is higher for More compared to Same trials? Or do a lagged regression (similar to Figure 4c, but now for behaviour) to show the latter samples have an impact on confidence but not performance on More trials?

I found the writing hard going in places. It was often difficult to figure out what exactly had been done in analysis – in particular "representation precision" (line 228) was only briefly defined in the figure legend, and it would be useful to spend some more time unpacking this in the text to help the reader follow along.

Line 399 of the supplement, "We used the representation error as an estimate of the inference error of the observer: the absolute difference between the cluster predicted value and the expected value given the cluster representation and the true value of the accumulated evidence based on the orientations presented to the observer." There are two "ands" here, so I did not understand what the absolute difference was between.

---

## [Author Response]

Essential Revisions (for the authors):1) The study needs to be more clearly framed with respect to the previous literature. Extensive neuroimaging and lesion work has already provided compelling evidence that perceptual decisions and metacognition rely on distinct neural circuits. In addition, previous modeling work that has provided a joint account of first- and second-order behaviour has assigned a key role to post-decision processing (whether within same or different circuits). The authors should clarify their hypotheses and how they relate to this previous work.

We thank the reviewers and the editor for highlighting this lack of clarity in the previous manuscript. We agree that the original introduction lacked sufficient detail with regards to the neuroimaging literature (relating to R1C1 and R2C2), and in what ways this experiment specifically adds to this literature. We have provided further details at P2L58 of the revised manuscript:

“Several experiments have linked modulations in confidence with activity in a variety of subregions of the prefrontal cortex (including the orbitofrontal cortex, Masset et al., 2020, Lak et al., 2014; right frontopolar cortex, Yokoyama et al., 2010; rostro-lateral prefrontal cortex, Fleming et al., 2012, Geurts et al., 2021; inferior frontal sulcus, medial frontal sulcus and medial frontal gyrus, Cortese et al., 2016; see also Vaccaro and Fleming, 2018, for a meta-analysis). Moreover, disrupting the processing in subregions of the prefrontal cortex (Rounis et al., 2010; Lak et al., 2014; Fleming et al., 2014) tends to impair (though not obliterate) the ability to appropriately adjust behavioural confidence responses, whilst leaving perceptual decision accuracy largely unaffected (though these results can be difficult to replicate, Bor et al., 2017; Lapate et al., 2020, and may not generalise to metacognition for memory; Fleming et al., 2014).”

And further at P3L72:

“One promising avenue of research for separating the mechanisms of metacognition from perceptual processes has been to utilise tasks where the observer may integrate additional evidence for confidence after they have committed to their perceptual decision (Murphy et al., 2015; Fleming et al., 2018), which presumably relies on processing independent of the perceptual decision. These studies show that post-decisional changes in confidence magnitude correlate with signals from the posterior medial frontal cortex.”

The original introduction also did not clearly disambiguate two key points (relating more specifically to R1C1) that a ‘general metacognitive mechanism’: 1) would reflect the computation of confidence (beyond the perceptual processes, i.e., not merely to transform confidence into a behavioural response), and 2) would be ‘separable’ in that the neural processes can be recruited independently of perceptual processes (allowing the evaluation of confidence across different forms of decision-making). We make this clearer at P2L36 and P2L66 respectively:

“At the neural level, perceptual confidence could therefore follow a strictly serial circuit: Relying only on information computed by perceptual processes, with any additional processes contributing only to transform this information for building the confidence response required by the task.

A challenge in this literature is in specifically relating the neural processing to the computation of confidence, as opposed to transforming confidence into a behavioural response, or a downstream effect of confidence, such as the positive valence (and sometimes reward expectation) accompanying correct decisions. Moreover, identifying how these neural mechanisms could be separable from the underlying perceptual processes is important for understanding the computational architecture of metacognition.”

In addition, Reviewer 2 highlighted the need for clarity with respect to the approach of comparing behaviour and EEG to an optimal observer, and how this relates to the computation of confidence. We have changed the terminology (in the introduction and the results) to make clear that we are not studying ‘confidence suboptimalities’ but rather, we are examining processing responsible for driving confidence to be greater or less than it should be given the presented evidence and the perceptual decision. This is activity that reflects the internal evidence the observer uses to make their confidence evaluations and therefore must be important for the neural computation of confidence. At P3L98 we re-phrase as:

“We find two distinct representations of the accumulated evidence. The first one reflects the internal evidence used to make perceptual decisions. The second representation reflects the internal evidence used to make confidence evaluations (separably from the perceptual evidence), and is localised to the superior parietal and orbitofrontal cortices.”

We hope that the additional analysis (see E2) also helps with this point.

2) Substantial rewriting of the Results is necessary. A clear rationale should be provided for each analysis step and the results of those analyses need to be linked back to the study hypotheses.

We thank the editor for this comment and the reviewers who have highlighted specific details of where the previous Results section was unclear. We have re-written and expanded the initial paragraph of each part of the Results section, to provide a clear rationale for each analysis step. We also include an overview at the beginning of the Results section, so the reader is given a clearer picture of the overall flow of the analysis steps and how they link to the hypothesis, P4L108:

“We present analyses to address two key hypotheses in this experiment: First, that observers are prematurely committing to their perceptual decisions whilst continuing to monitor additional evidence for evaluating their confidence. And second, that there are separable neural signatures of the evaluation of confidence during perceptual decision-making. To address the first hypothesis, we use a combination of behavioural analyses and computational modelling, and in addition, show that the EEG signatures of response preparation are triggered from the time of decision commitment, even when this occurs seconds prior to the response cue. To address the second hypothesis, we use the stimulus evoked responses in EEG to trace the representation of the presented evidence throughout each trial. We show that these neural representations of the optimal accumulated decision evidence are less precise when the observers’ behavioural responses were also less precise relative to optimal. We use this to isolate clusters of activity that specifically reflect the internal evidence used for observers’ confidence evaluations beyond the presented evidence. We then localise the sources of this activity, and relate these processes back to observers’ eventual confidence ratings.”

We also looked out for where our descriptions of the analyses were unclear, as highlighted more specifically by the Reviewers comments.

3) Consider whether a simpler model-free analysis (e.g. check whether confidence resolution is higher for More vs Less trials) could be done to validate the key claim of confidence/performance dissociations (see Reviewer 2 comment)

We thank the reviewer and the editor for this suggestion, which has strengthened the evidence in support of the computational dissociation between the evidence used for perceptual and confidence responses. We’ve included a similar GLM as used to compare the neural signature of response execution and the presented evidence, as suggested by Reviewer 2 (R2C3). We note here that this analysis is not entirely ‘model-free’, as it relies on the definition of the optimal observer to compute the evidence in favour of the presented evidence, and on the assumptions of the GLM (that the variables are related by a probit link function). However, it is difficult to assess confidence without some assumption of how confidence should vary based on the presented evidence. Whilst in other experiments there is often a specific external manipulation of task difficulty, here we manipulate the number of samples, which does not directly map on to task difficulty (three samples at 45 degrees is much better than three vertical samples, and indeed, observers’ reported confidence does not systematically vary with the number of samples, see Appendix 3), hence the need to compute the evidence of the optimal observer. We hope that the reader is persuaded that this assumption is not invalid, given that the optimal evidence is a strong predictor of observers’ responses in this task (including the confidence responses, see Appendix 3), and the substantial previous work examining how behaviour varies with the optimal evidence in this kind of task (Drugowitsch et al., 2016). This analysis is presented on P6L198:

These modelling results are supported by an analysis using general linear models to examine the relationship between the optimal presented evidence, *L*, and observers’ behaviour in the perceptual decision and confidence evaluation. As stated above, *L* is the evidence that which maximises the probability of a correct response: the accumulated difference in the log probabilities of the presented orientations given the category distribution (Figure 1e). First, we find the presented evidence accumulated over all samples does explain substantial variance in observers’ perceptual decisions (average 𝛽= 0.77, *t*(19) = 6.48, p < 0.001), and confidence evaluations (with the evidence signed by the perceptual response; 𝛽= 0.24, t(19) = 6.46, p < 0.001). This suggests that the internal evidence that observers were using to make their responses, *L**, correlated significantly with the optimal evidence *L* (as has been found previously; Drugowitsch et al., 2016). Second, the total accumulated evidence in the More condition was not a significantly better predictor of the observers’ perceptual decisions than the evidence up to four samples prior to the response (average difference in 𝛽 = 0.034, *t*(19) = 1.63, p = 0.12), while for the Same and Less conditions the total accumulated evidence was a significantly better predictor (Less: *t*(19) = 4.99, p < 0.001; Same: *t*(19) = 3.11, p = 0.006; causing a significant interaction between condition and sample accumulated to *F*(2,38) = 10.348, p = 0.001, Bonferroni corrected for three comparisons, Figure 2f, top). This supports the finding from model comparison and behaviour that observers implemented a covert bound on perceptual evidence accumulation. And finally, this interaction was not present when examining how the presented evidence affected confidence evaluations (*F*(2,38) = 3.124, p = 0.09, uncorrected, Figure 2f, bottom). Rather, the accumulated evidence up to the final sample in the More condition was a significantly better predictor of confidence than the evidence accumulated to four samples from the response (average difference in 𝛽 = 0.26, *t*(19) = 5.33, p < 0.001), supporting the prediction from the computational model analysis that observers integrated all the presented evidence for evaluating confidence.

4) Check impact of high-pass filtering and current source density on CPP signals (or consider removing those signals from the analysis if they are not speaking to the study hypotheses).

We thank the Editor and Reviewer 1 for this comment (R1C3), which has improved the CPP analysis. We removed the high-pass filter and applied the current source density transformation, we also used the same approach as in Kelly and O’Connell, 2013, in comparing the slope of the CPP in a pre-response window. The main comparisons showed similar results as before: no significant effect of condition (now analysed across all three conditions), a significant effect of perceptual decision accuracy, and a significant effect of confidence magnitude. However, we decided to move this analysis to the Supplementary Materials (now called Appendix 4), as it does not substantially contribute to the main hypotheses, though we consider the CPP an important part of the literature on perceptual evidence accumulation and confidence.

Reviewer #1 (Recommendations for the authors):Some substantial rewriting of the Results section is required to clarify the rationale for each analysis without requiring that the reader read all of the Supplemental Materials in order to follow the narrative. In several instances it is not clear why the analysis is being run or what we should take from it. A prominent example is the ERP analysis where the choice of signals or their relevance to the hypothesis is not explained and the relevance of the reported results to those hypotheses is similarly not clarified. It is not clear why only the More and Less conditions are compared or why post-response time-points are considered relevant.

Since the ERP analysis was not central to our arguments, we have moved this analysis to the Supplementary materials (now called Appendix, more details in E4).

Additionally the purpose of the decoding analysis is left obscure and it is not clear why 8-32Hz is the focus

We have now included more detail to motivate this analysis (P8L242):

“The analysis of behaviour and computational modelling so far has suggested that observers were committing to their perceptual decisions early in the More condition and ignoring the additional evidence for their perceptual decision. We questioned the extent of this covert decision commitment, that is, whether observers were going as far as to plan their motor response before the response cue.”

And more detail about the choice of methods (P8L246):

“Initial analysis suggested the spectral power in the 8 to 32 Hz frequency range (the ‘α’ and ‘β’ bands) could be used to classify perceptual decisions based on lateralised differences over motor cortex (Appendix 5). A classifier was trained to discriminate observers’ perceptual decisions at each time-point in a four second window around the response in the Free task (3 seconds prior to 1 second after).”

One of the key observations is that the precision of neural representations of accumulated evidence drop towards the end of the More trials but there is no such dropoff in the decision update representation. I am struggling to understand how this might arise and it would be helpful if the authors could provide an explanation.

We note that this was originally a point in the discussion (P15L458):

“This selective dampening of the representation of accumulated evidence following premature decision commitment delineates which computations are devoted solely to the perceptual decision process, and which computations reflect the input to the decision process: The representations of stimulus orientation and decision update (Wyart et al., 2012; Wyart et al., 2015; Weiss et al., 2021), which are necessary input for the perceptual decision, did not substantially change after committing to a perceptual decision. This initial perceptual processing stage likely remained important for the continued accumulation of evidence for evaluating confidence (even after the completion of perceptual decision processes), though it could also be that these processes are automatically triggered by stimulus onset irrespective of whether the evidence is being accumulated for decision-making.”

We now also cue this important point in the Results section (P11L329):

“These differences in representation precision were not present for the encoding of stimulus orientation, nor the decision update, suggesting that these processes may reflect input to perceptual evidence accumulation, but not the accumulation process itself.”

The authors report that there is no significant effect of More vs Same on choice accuracy but the figure suggests that there is a substantial numerical difference in the expected direction. Bayes factors should be provided to quantify the evidence in favour of the null hypothesis.

We have rephrased the paragraph to better highlight that there is an average increase in sensitivity in the More condition (P5L164):

“Replicating these previous results (Balsdon et al., 2020), we found that perceptual decision sensitivity (d’) was significantly decreased with just two fewer stimuli in the Less condition compared to those same (pmin) trials in the Free task (Wilcoxon sign rank Z = 3.88, p < 0.001, Bonferroni corrected for three comparisons, Figure 1a), but four additional stimuli (Figure 1b) in the More condition resulted in only a small but not significant increase compared to the pmax trials in the Free task (Z = -1.53, p = 0.13, uncorrected). There was also no significant difference for the Same condition (Z = 1.21, p = 0.23, uncorrected; Figure 2a).”

We note that our model (with the covert bound) also predicts this increase (red circle in Figure 2a). The important aspect of the data is not that there is no increase in sensitivity, but that sensitivity is better predicted by a covert bound than by the predicted sensitivity from Same and Less condition (the significant result from the model comparison). In addition, thanks to the suggestion in particular from Reviewer 2 (R2C3), we now show a significant difference for the More condition: We compare how well perceptual decisions are predicted from the presented evidence accumulated up to the last sample and the fourth from last sample, and show a significant interaction between the sample accumulated to and the condition (whilst the responses of the Same and Less conditions are better predicted by the evidence accumulated to the last sample, the responses in the More condition are not).